# Comparative ozone production sensitivity to $NO_x$ and VOCs in Quito, Ecuador and Santiago, Chile

María Cazorla[1], Melissa Trujillo[1], Rodrigo Seguel[2,3], Laura Gallardo[2,3]

[1]Universidad San Francisco de Quito USFQ, Instituto de Investigaciones Atmosféricas, Quito, Ecuador
[2]Center for Climate and Resilience Research (CR2), Universidad de Chile, Santiago, Chile
[3]Departamento de Geofísica, Facultad de Ciencias Físicas y Matemáticas, Universidad de Chile, Santiago, Chile

*Correspondence to*: María Cazorla (mcazorla@usfq.edu.ec)

**Abstract**. Amid the current climate and environmental crises, cities are being called to reduce levels of atmospheric pollutants that also act as short-lived climate forcers, such as ozone and $PM_{2.5}$. This endeavor presents new challenges, especially in understudied regions. Here, we use a chemical box model to investigate the ozone production sensitivity to $NO_x$ and VOCs in Quito, Ecuador and Santiago, Chile. We present ozone production rates ($P(O_3)$) calculated using VOC measurements taken in Santiago, along with VOC vs. CO linear regressions (LRs), and complementing the analysis with Monte Carlo (MC) simulations. In Quito, VOC measurements are unavailable for which we simulated a range of concentrations using LRs and MC simulations. We modeled $P(O_3)$ in March 2021 and for typical conditions per season in 2022. We calculated a range of $P(O_3)$ in Quito of 15-50 ppbv $h^{-1}$ year-round. In Santiago, we found that $P(O_3)$ is 23-50 ppbv $h^{-1}$ in the ozone season (austral summer). Although the $P(O_3)$ magnitudes were found to be comparable, Santiago has a well-established ozone season, unlike Quito, where concentrations are lower. From sensitivity experiments, alkenes and aromatics contribute 50% to $P(O_3)$ in Santiago and could reach 70-90% in Quito (noon and afternoon). Aldehydes and ketones contribute 30-40% in Santiago and about 20% in Quito (noon and afternoon). We estimate the isoprene contribution to be 20% in Santiago and 10% in Quito. VOC reduction experiments generally lowered $P(O_3)$ in both cities. In Santiago, $NO_x$ reductions increased the morning $P(O_3)$.

## 1. Introduction

Tropospheric ozone bears a dual nature of a serious air contaminant and a short-lived climate forcer (SLCF). At the ground level, ozone negatively impacts human health and vegetation due to its oxidative nature that affects the respiration function in living beings (Fleming et al., 2018; Gaudel et al., 2018; Karlsson et al., 2017; Malley et al., 2017; Mills et al., 2018; Soares and Silva, 2022). Furthermore, tropospheric ozone is the third largest anthropogenic climate forcer (Anon, 2014; Checa-Garcia et al., 2018; Skeie et al., 2020), and it deters carbon sinks as it is a limiting factor to carbon capture in vegetation (Mills et al., 2016; Zhang et al., 2022). Hence, mitigating the ozone abundance in the ambient air is an action that simultaneously protects public health while it combats climate change. In most cities across the world, ozone is considered one of the criteria air pollutants and thus is regulated by local and national legislation (Lyu et al., 2023). However, ozone continues to be a major air quality concern in many regions of the world despite decades of studying the complex and non-linear nature of its production chemistry. This complexity imposes tailoring control strategies appropriate for each city that carefully consider the chemical composition of the ambient air.

The mitigation of ozone to benefit in parallel air quality and climate presents a new challenge in devising efficient controls on ozone precursors at the urban level. This need is particularly pressing in understudied regions that are highly vulnerable to

climate change and that still struggle with poor air quality, such as cities in South America (SA) (Cazorla et al., 2022). In this work, we investigate the sensitivity of ozone production to its precursors ($NO_x$ and VOCs) in a comparative fashion between Santiago (Chile) and Quito (Ecuador) during time periods in 2021 and 2022. To this end, we use a constrained photochemical box model, namely the F0AM (Framework for 0-D Atmospheric Modeling) (Wolfe et al., 2016). We present ozone production rates ($P(O_3)$) calculated using VOC measurements taken in Santiago, along with VOC vs. CO linear regressions (LRs), and

complementing the analysis with Monte Carlo (MC) simulations. In Quito, in situ measurements of VOCs are not available. Hence, we used MC simulations to generate an array of VOC inputs to the model. We present the results as a series of sensitivity model runs. We discuss the impact of changing the precursor proportion on the chemistry of ozone production in both cities.

A recent ozone trend study in South America demonstrated that the short- and long-term ozone exposure in tropical cities Quito

and Bogota are lower than those encountered at extratropical cities Santiago and São Paulo (Seguel et al., 2024). This regional distribution of ozone exposure seems somewhat counterintuitive, given year-round solar radiation at high altitudes over tropical Andean cities combined with their intense traffic emissions. Previous work showed that a VOC-limited environment constrained ozone production in Quito, but rates of ozone production were not compared to other cities in the region (Cazorla, 2016). On the other hand, Santiago has been dealing with over two decades of ozone pollution and seasonal exceedances that are worsening in

time due to an increased frequency of extreme heat and wildfire events (Seguel et al., 2020, 2024). Here, we compare the ozone production chemistry between both cities to gain insight into mechanisms that can lead to ozone formation in the ambient air.

An important angle to consider is the effect that a shift in precursors would have on ozone production rates. For example, many cities in SA, including Quito and Santiago, have a seasonal high $PM_{2.5}$ problem (Gómez Peláez et al., 2020). As with ozone,

reducing levels of $PM_{2.5}$ in cities protects public health while it is an effective climate action (Szopa et al., 2023). A direct way of cutting down levels of $PM_{2.5}$ is reducing diesel-based traffic emissions, which in turn lowers $NO_x$ levels. This beneficial outcome was observed during the COVID-19 pandemic confinements, when primary pollutants and $PM_{2.5}$ decreased immediately as a direct consequence of mobility restrictions. However, the effect on ozone was the opposite. Research conducted in Quito, Santiago and other South American cities showed that $NO_x$ reductions increased ozone production rates and ground-level ozone

due to a shift in the production chemistry from a VOC-limited towards a more $NO_x$-limited regime (Cazorla et al., 2021; Seguel et al., 2022; Sokhi et al., 2021). Meanwhile, in the free troposphere ozone generally decreased (Putero et al., 2023). Another example worth-noting is a modeling study in the city of Cuenca, Ecuador, that showed how the sole measure of replacing the diesel-based public transportation by electric vehicles would decrease $PM_{2.5}$ and $NO_2$, but the levels of ambient ozone would increase (Parra and Espinoza, 2020).  These results impose a need to assess the impact of shifting the composition of precursors

on the ozone forming chemistry that could arrive from applying controls on individual contaminants. It follows that assessing combined changes in both sets of precursors ($NO_x$ and VOCs) is critical, which we present in this work.

With the above motivations, we employ a sensitivity approach to address the following research questions:

a)   How do the typical magnitudes of ozone production rates compare between Quito and Santiago?
          b)   What chemical groups of VOCs contribute the most to ozone production in each city?
          c)   What would be the effect on ozone production if drastic reductions in $NO_x$, VOCs or both were applied in each city?

In the Methods section, we briefly describe ozone chemistry and the way we calculate ozone production and loss rates as well as radical production rates. Furthermore, we provide details of the study sites, time periods, and data sets. In addition, we give a full explanation of model settings, constraints, and runs. In the Results and Discussion section, we compare pollutant levels, magnitudes of ozone production rates, radical abundances, and radical production rates between both cities. In addition, we quantify the contribution of different VOC groups to ozone production, and we discuss ozone production rates under different sensitivity scenarios of $NO_x$ and VOCs. In the Conclusions section we present the main findings.

## 2. Methods

### 2.1 Ozone and radical production and losses

The abundance of ozone in the ambient air is the result from a balance between chemical production, chemical loss, dry deposition, and ozone transport due to advection (Seinfeld and Pandis, 2016). Sometimes, stratospheric intrusions could also contribute to the ozone budget (Archibald et al., 2020). The net chemical production rate, denoted as $P(O_3)$ in this study (instantaneous chemical production minus chemical loss), is a direct source of ozone in the ambient air. Under stable atmospheric conditions, $P(O_3)$ is the main factor that causes ozone accumulation within the mixing layer. The chemistry of ozone production has been studied extensively for several decades (Gery et al., 1989; Haagen-Smit and Fox, 1956; Kleinman, 2005a; Logan et al., 1981; Thornton et al., 2002a). A simplified mechanism is depicted in reactions R1-R8. The hydroxyl radical, OH, oxidizes VOCs and leads to the formation of the hydroperoxyl radical, $HO_2$, and other peroxy- radicals, $RO_2$ (R1). The latter two react with fresh emissions of NO, which produces $NO_2$ (R2, R3). From these reactions, OH and other organic radicals (RO') are formed. The photolysis of $NO_2$ with daylight splits the molecule into ground state oxygen, O, and reforms NO (R4). Subsequently, O rapidly reacts with $O_2$ and ozone is formed (R5). Depending on the proportion of VOCs to $NO_x$, both sets of precursors compete to react with OH. Hence, OH and $HO_2$ radicals very rapidly cycle in a catalytic fashion to feed the mechanism with new $NO_2$ that undergoes photolysis and forms ozone. The titration of NO by ozone (R6) is not a real ozone sink during daytime due to the rapid $NO_2$ photolysis. In contrast, reactions that consume radicals such as the formation of nitric acid (R7) or the formation of hydrogen peroxide (R8) lead to chain termination. From this mechanism, the rate equation for the instantaneous chemical production of ozone (p) is derived from reactions R2 and R3, as is depicted in equation (1) (Ren et al., 2013; Seinfeld and Pandis, 2016; Shirley et al., 2006; Thornton et al., 2002b).

| | |
|---|---|
| $OH + VOC \rightarrow HO_2, RO_2 + H_2O + O_2$ | (R1) |
| $HO_2 + NO \rightarrow NO_2 + OH$ | (R2) |
| $RO_2 + NO \rightarrow NO_2 + RO'$ | (R3) |
| $NO_2 + h\nu \rightarrow O + NO$ | (R4) |
| $O + O_2 + M \rightarrow O_3 + M$ | (R5) |
| $O_3 + NO \rightarrow NO_2 + O_2$ | (R6) |
| $OH + NO_2 + M \rightarrow HNO_3 + M$ | (R7) |
| $HO_2 + HO_2 \rightarrow H_2O_2 + O_2$ | (R8) |

$$p = k_{HO2+NO}[NO][HO_2] + \sum k_i[NO][RO_2]_i \quad\quad (1)$$

Meanwhile, ozone loss is driven by reactions that deplete radicals. In urban environments, the reaction of OH and $NO_2$ to form nitric acid (R7) is a main mechanism of ozone loss (Seinfeld and Pandis, 2016; Sillman, 1995; Thornton et al., 2002b). Other

ozone losses are $O_3$ photolysis, the reaction of $O_3$ with $HO_2$, the reaction of alkenes (ALK) and $O_3$, and the reaction of $RO_2$ radicals with NO to produce organic nitrate, $P(RONO_2)$. Thus, the net ozone production rate, $P(O_3)$, can be calculated using equation (2), where $k_i$ represents reaction rate constants, $J_{O3}$ is the frequency of photolysis for ozone (R9), and species abundance is depicted within brackets.

$$P(O_3) = k_{HO2+NO}[NO][HO_2] + \sum(k_i[NO][RO]_i) - k_{OH+NO2+M}[OH][NO_2][M] - J_{O3}[O_3] - k_{HO2+O3}[HO_2][O_3] - \sum(k_i[O_3][ALK]_i) - P(RONO_2) \tag{2}$$

In a VOC-limited regime, $HO_x$ radicals are lost due to reactions with $NO_x$ with R7 being a well-known chemical fate (Kleinman, 2005a; Kleinman et al., 2001a; Sillman, 1995). In addition, the production of organic nitrate can also be an important loss (Schroeder et al., 2017). In a $NO_x$-limited regime, reactions between $HO_x$ radicals (collectively OH and $HO_2$) have been identified as an important chemical loss. For example, the reaction between two $HO_2$ radicals produces hydrogen peroxide, $H_2O_2$ (R8). The equations to quantify these two radical losses are depicted as $L_{NOx}$ and $L_{ROx}$ in equations (3) and (4).

$$L_{NOx} = k_{OH+NO2+M}[OH][NO_2][M] + P(RONO_2) \tag{3}$$

$$L_{ROx} = 2k_{HO2+HO2}[HO_2]^2 \tag{4}$$

In this work, we use the ratio $L_{NOX}/(L_{NOX}+L_{ROX})$ as an indicator of the chemical regime of ozone production (Kleinman, 2005b; Kleinman et al., 2001b; Schroeder et al., 2017). With this method, ratios greater than 0.5 indicate that radical losses to the production of nitric acid dominate, which takes place in a VOC-limited ($NO_x$-saturated) regime, while ratios below 0.5 indicate that reactions among radicals, such as R8, are important and mark a shift towards the $NO_x$-limited regime. According to (Schroeder et al., 2017), the 0.5 threshold should be revisited to 0.74 (or $L_{ROx}/L_{NOx} = 0.35$) by incorporating losses to the production of organic nitrate. Other useful indicators include the ratio of formaldehyde to reactive nitrogen $HCHO/NO_y$ for which a ratio below 1 is an indicative of a VOC-limited regime (Souri et al., 2020). Here, we used model output of this ratio to complement the analysis.

With respect to the production of $HO_x$, we examine the main atmospheric sources of OH and $HO_2$ (independently from reactions R2 and R3, which are discussed in the ozone production section) to calculate radical production rates, $P(HO_x)$. Thus, reactions (R9) to (R11) briefly depict ozone photolysis followed by the reaction of $O^1D$ with water vapor, which produces OH radicals (Levy, 1971). Other important sources of OH and $HO_2$ in the urban atmosphere are the photolysis of HONO and formaldehyde (R12 and R13) (Dusanter et al., 2009; Ren, 2003; Ren et al., 2013). Moreover, the reaction of ozone and alkenes (ALK) is known to produce OH and other radicals (R14). Therefore, equations (5) to (8) were used to quantify these rates.

$$O_3 + h\nu \rightarrow O_2 + O^1D \tag{R9}$$

$$O^1D + M \rightarrow O + M \tag{R10}$$

$$O^1D + H_2O \rightarrow 2OH \tag{R11}$$

$$HONO + h\nu \rightarrow OH + NO \tag{R12}$$

$$HCHO + h\nu \rightarrow 2HO_2 + CO \tag{R13}$$

$$ALK + O_3 \rightarrow OH + RO' \tag{R14}$$

$$P(HO_x)_1 = 2J_{O_3 \to O1D}[O_3]\, k_{(O1D + H2O)}\, [H_2O]/(k_{(O1D + M)}[M]) \tag{5}$$

$$P(HO_x)_2 = 2J_{HCHO}[HCHO] \tag{6}$$

$$P(HO_x)_3 = J_{HONO}[HONO] \tag{7}$$

$$P(HO_x)_4 = \sum(k_i[O_3][ALK]_i) \tag{8}$$

## 2.2 Study sites and background information

A map with the location of the two study sites in Quito, Ecuador, and Santiago, Chile is shown in Fig. 1. Details of the stations chosen for the study as well as background information follow.

One of the sites is located at Universidad San Francisco de Quito's Atmospheric Measurement Station (EMA USFQ, Spanish acronym) at coordinates 0.19°S, 78.43°W, and 2414 m a.s.l. Quito (2.78 million inhabitants) is an Andean city located at high altitude on the equator. In Quito, the typical UV index is greater than 11 on 40.0–76.1% of days per month, while the number of days per month with UV index greater than 16.0 varies between 0.7% and 32.0% (Parra et al., 2019). Particularly, intense insolation and UV index take place during the equinox months of March and September. In parallel, the times of the year with increased precipitation peak around March-April and November, while the drier months run from July to September (Cazorla et al., 2024), although conditions vary mildly within a year. A discussion on weather patterns that influence air quality in Quito can be found in previous work (Cazorla and Juncosa, 2018).

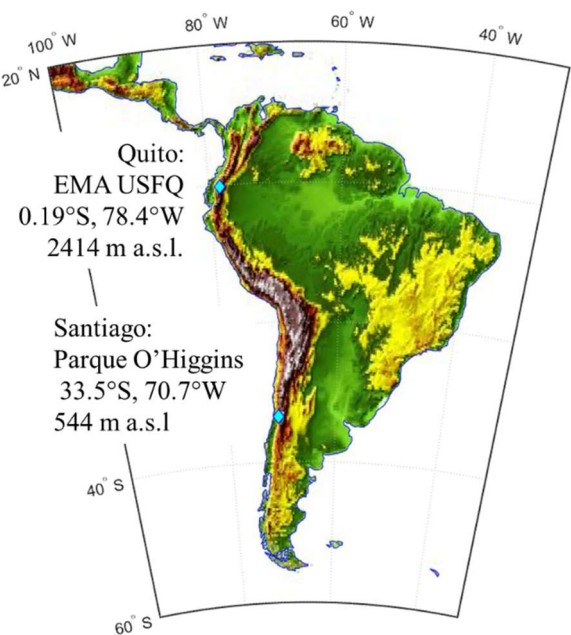

**Figure 1.** Map of South America with location and coordinates of EMA USFQ station in Quito, Ecuador and Parque O'Higgins station in Santiago, Chile.

In Santiago, we chose Parque O'Higgins station, located some 5 km away from downtown Santiago, at coordinates 33.46°S, 70.66°W, and 535 m a.s.l. This station is run by the Ministry of the Environment, and it typically represents the city's average conditions (Osses et al., 2013). Santiago (8 million inhabitants) is a subtropical city that exhibits four seasons. The warmest and sunniest months take place during the austral summer (December to February), but the high ozone season often extends until

March. This is an interesting month in terms of air quality in Santiago because high temperatures and insolation prevail, while anthropogenic emissions and ozone precursors increase as the school year starts and the flow of people into the city increases.

## 2.3 Study time periods

### 2.3.1 Santiago

Ozone production was calculated for time periods in 2021 and 2022. A field campaign to measure VOCs in Santiago was conducted in March 2021 (description in section 2.4.2 and Appendix A). Therefore, we performed detailed modeling of the chemistry of ozone production for sunny days during the month of March. We filtered out overcast days because photolysis frequencies were not measured and, thus, we rely on modeled actinic flux without cloud cover correction for the photolysis component of the model. March 2021 was mostly sunny in Santiago, so only 5 days were filtered out. Fig. S1 in the Supplement shows the time series of solar radiation overlapping mostly sunny days to all days.

To further explore ozone production under additional precursor conditions that are typically observed within a year, we run simulations for average conditions per season in 2022. Thus, we run the model for a typical sunny day in austral summer (February and March, data in January was not included because of poor quality), fall (April, May), winter (June to August), and spring (September to November). As in the previous case, we filtered out overcast days from the time series. Mean conditions were found by overlapping meteorological and air quality data (1-hour resolution) in 24-hour plots and finding the average for every hour, namely the mean diurnal variation (MDV). These mean conditions were used as inputs to run the model. Model input variables are described in section 3.

### 2.3.2 Quito

As in the case of Santiago, we performed detailed photochemical modeling for sunny days in March 2021 (11 overcast days were filtered out, Figure S1) and for typical sunny conditions in 2022. Thus, we run the model for typical conditions in January, often a sunny month; June-July, a time with low precursors due to summer vacation; September, an equinox month with increased precursors due to the return from summer vacation; and October-November, a time of transitioning into the rainy season. Mean conditions (sunny days only) for each period were found in a similar way as it was done for Santiago.

## 2.4 Data

### 2.4.1 Ozone, NO, $NO_2$, CO, and meteorological data

*Santiago*

Air quality data and meteorological conditions (1-hour) measured at Parque O'Higgins station were obtained from the Air Quality National Information System maintained by the Ministry of Environment (SINCA, Spanish acronym, https://sinca.mma.gob.cl/). From network information, ozone is measured with UV photometry, $NO_x$ with chemiluminescence, CO with infrared photometry, and published data complies with quality control standards according to national legislation (https://sinca.mma.gob.cl/index.php/documentos). Fig. S2 (Supplement) depicts the time series of measurements in March 2021. For comparisons with Quito, the time coordinates in all figures in this work are presented in UTC-5 (local and solar time in Quito and roughly solar time in Santiago).

All data described in this section are available as complete input files for the F0AM model for Quito (Cazorla et al., 2025a) and Santiago (Cazorla et al., 2025b), see also the Data Availability statement.

*Quito*

Ozone is measured at EMA USFQ with a Thermo 49i UV photometer. Measurements are periodically intercompared against a 2B-Technologies UV photometer. The agreement between measurements is better than 5%. A Teledyne 400 chemiluminescence instrument is used to measure $NO-NO_2-NO_x$. Calibration is done with a certified NO standard and zero air to prepare calibration mixtures. Uncertainty in measurements is better than 5% (Cazorla, 2016). The original rate of acquisition of ozone and $NO_x$ measurements is 1-second. Meteorological measurements are also available at EMA USFQ with an original rate of acquisition of 30-second. CO is not measured at EMA USFQ. However, the car fleet is similar in the entire city. Hence, following previous work (Cazorla et al., 2021), we used an average of CO measurements from three stations (Belisario, Centro and Tumbaco) run by the Quito Air Quality Network (Secretariat of the Environment, Quito, Ecuador) (https://aireambiente.quito.gob.ec/). Fig. S3 depicts the time series of Quito observations.

**2.4.2 In situ measurements of VOCs in Santiago**

VOC measurements in Santiago were taken from a fourth-story window at the Geophysics Department of Universidad de Chile (33.457 ºS, 70.662 ºW, 535 m a.s.l.) from 17 to 28 March 2021, corresponding to the late summer. The site is located at campus Beauchef, in downtown Santiago, 770 m away from Parque O'Higgins station, where ozone, carbon monoxide, nitrogen oxides, and meteorological parameters are measured.

VOCs were measured using PTR-TOF-MS (proton transfer reaction time of flight mass spectrometry) at 1-second resolution. The instrument used was an Ionicon Analytik GmbH, Model 1000. A detailed description of the sampling operation of the PTR-TOF-MS instrument, calibration and mass discrimination methodology as well as the limit of detection of measured compounds is given in Appendix A. Table 1 shows the VOCs utilized in this study. Grouping was done for the practical purpose of organizing model runs, as explained in Section 2.5. The ISO nomenclature contains isoprene and monoterpenes also for practical reasons (isoprene contributes to $P(O_3)$ formation but the monoterpene contribution was found to be marginal). In Appendix B (Fig. B1), we present the time series of measured VOCs. Based on details in Appendix A, we estimate a conservative error of 30-40% in all VOC observations.

Previous studies conducted in numerous cities demonstrate that ambient measurements of VOCs, especially those whose source is fuel in vehicles, linearly correlate with CO, an anthropogenic tracer of combustion (Baker et al., 2008; Bon et al., 2011; Borbon et al., 2013; Brito et al., 2015). These correlations are useful as they can be used as emission ratios (Borbon et al., 2013). Thus, we obtained linear regressions between VOC and CO observations in Santiago for the campaign period. To this end, we linearly correlated the diurnal cycle of each VOC with the diurnal cycle of CO. We checked the results by correlating nighttime VOC and CO measurements and subtracting the background CO (noontime concentration) from the time series (Borbon et al., 2013; Brito et al., 2015). Both approaches yielded about similar slope values, but the determination coefficients were generally lower for the second approach. However, the determination coefficients improved for cresol, acetaldehyde, acetone, and isoprene. Thus, only for the latter compounds we used the linear fits obtained with the second approach. Table 1 shows the linear fit parameters used in this work. Table S1 contains the parameter regressions obtained with both approaches. Graphs for the VOC vs CO regressions for benzene and toluene can be found in Fig. S4 and S5. The entire compendium of regression figures (both methods) can be found in the link indicated in the Data Availability statement.

Compared to VOC/CO ratios (slope of linear fit in pptv/ppbv) reported for other cities in Latin America, namely Mexico City (Bon et al., 2011) and São Paulo (Brito et al., 2015), the slope we found for benzene (2.57) is about twice the value reported for Mexico City (1.21) and 2.5 times the value for São Paulo (1.03). For toluene, the slope in this study (7.54) is also higher than in Mexico City (4.2) and São Paulo (3.1). For C9 aromatics, we found a value of 3.64, close to what was reported in Mexico City (2.8). The Mexico City and São Paulo studies were done about a decade ago. Further work is needed regionally to better compare these ratios across cities, preferably with more recent measurements.

Linear fits for compounds associated with traffic (10 cases) show $R^2>0.83$ and correspond to alkenes, aromatics, and some oxygenated compounds (methanol, ethanol, and phenol). Some of the latter compounds could also have a biogenic origin, but finding exact proportions from each source was not within the scope of this study. The determination coefficient is less good for other oxygenated compounds, namely acetaldehyde, acetone, and acetic acid ($R^2 \sim 0.61$), which can be attributed to the secondary nature of these compounds. As expected, the linear fit for isoprene is less good ($R^2 = 0.56$) due to its combined anthropogenic and biogenic origin. In contrast, monoterpenes correlate well ($R^2=0.82$), for which we associate these compounds with traffic sources in addition to biogenic sources as observed in other cities (Borbon et al., 2023).

**Table 1**. VOCs measured in Santiago from 17 to 28 March 2021 and linear regressions of VOCs vs. CO in ppbv.

| Group | Compound | VOC regressions (ppbv) | | |
| --- | --- | --- | --- | --- |
| | | Slope x $10^{-3}$ | Intercept | $R^2$ |
| ALK | Propene | 7.46 | -2.467 | 0.87 |
| | Butene | 9.66 | -3.07 | 0.84 |
| ARO | Benzene | 2.57 | -0.85 | 0.86 |
| | Toluene | 7.54 | -2.95 | 0.9 |
| | Ethylbenzene | 15.73 | -6.44 | 0.83 |
| | Styrene | 0.96 | -0.24 | 0.87 |
| | C9 Aromatics | 3.64 | -1.47 | 0.9 |
| OXY | Methanol | 19.59 | -5.10 | 0.87 |
| | Ethanol | 4.87 | -2.23 | 0.87 |
| | Phenol | 0.98 | -0.16 | 0.84 |
| | Cresol | 0.29 | 0.27 | 0.45 |
| ALD | Acetaldehyde | 8.05 | 3.35 | 0.62 |
| | Acetone/Propanal | 8.57 | 3.33 | 0.61 |
| | Butanone/Butanal | 4.70 | -1.30 | 0.73 |
| | Methacrolein/MVK | 0.88 | 0.21 | 0.74 |
| | Acetic acid/Glycolaldehyde | 14.70 | -2.06 | 0.59 |
| ISO | Isoprene | 0.88 | 0.58 | 0.56 |
| | Monoterpenes | 0.66 | -0.14 | 0.82 |

**2.4.3 Monte Carlo perturbations on VOCs in Santiago**

Direct measurements of VOCs were not available in Santiago on all days of March 2021 or on typical days per season in 2022. In addition, the available measurements were subject to methodological limitations, as previously discussed. Given the error in measurements and the uncertainty of applying regressions in Table 1 to derive VOCs at times other than the campaign period, we ran ten thousand Monte Carlo (MC) simulations within +/- 35% of the known values to find an array of inputs to the F0AM

model. Thus, we ran individual MC simulations for 22 compounds (Table 1). From the bulk of each MC simulation, we extracted percentiles 10 to 90, which covered the simulation range. Fig. S6 depicts the flow diagram followed to generate MC simulations. In Fig. S7, we present the MC simulations for benzene (time series and percentile distribution as diurnal cycles). The entire compendium of MC simulation figures for all VOCs are accessible through the link provided under Data Availability.

### 2.4.4 Simulations of VOCs in Quito

Measurements of VOCs are unavailable in Quito. For this reason, we used a Monte Carlo approach to simulate VOCs and to find an array of inputs to the model. VOCs in Quito could be derived from CO provided suitable emission ratios were available (Borbon et al., 2013). Given the lack of experimental data, we generated VOCs using CO measurements in Quito and the regressions in Table 1. Subsequently, we perturbed these data ten thousand times within +/-50% through Monte Carlo simulations. This conservative factor was applied based on the observation that CO in Quito varies from CO in Santiago in about +/- 25% at noon, but during rush hours this variability could be up to +/-50%. From the bulk of the simulations, we extracted percentiles 10 to 90. This process was applied to all time periods analyzed in this study. In Fig. S8, we present the flow diagram followed to generate VOCs.

In Quito, 98% of ambient CO is a primary emission that comes from on-road traffic (Hernandez and Mendez, 2020; Parra, 2017; Vega et al., 2015). Hence, alkene and aromatic compounds such as propene, butene, benzene, toluene, xylenes as well as C8 and C9 aromatics are expected in the Quito environment. Fuel options such as gasoline mixtures with ethanol are not currently commercialized in Quito, but they are available in some other cities, mainly in the coast of Ecuador. Thus, we included ethanol and methanol to investigate the potential effect if these types of fuels were available. In addition, methanol can also be present from biogenic sources. Aldehydes and ketones are usually present in an urban environment as they are byproducts of oxidation and photolysis reactions.

We present MC simulations for benzene in Fig. S9. The entire compendium of MC simulation figures for all VOCs for Quito can be found in the link provided under Data Availability.

### 2.5 Model details

We used the F0AM (Wolfe et al., 2016) to model ozone formation chemistry in Quito and Santiago in the time periods indicated in Section 2.3. The chemical mechanistic information was taken from the Master Chemical Mechanism, MCM v3.3.1 (Bloss et al., 2005; Jenkin et al., 1997, 2003, 2015; Saunders et al., 2003), via website: www.mcm.york.ac.uk. The F0AM is a model that runs in MATLAB (https://la.mathworks.com/). Details of model settings, and simulations follow.

### 2.5.1 Model input and options

*Time step*

The time step for the March 2021 model runs for Santiago and Quito was 10 minutes. To this end, the original Quito data were integrated into 10-minute resolution. The same was done with Santiago 1-minute VOC data. Regarding air quality and meteorological observations in Santiago, 1-hour data were interpolated to generate 10-minute time series. In contrast, the 2022 runs were done using 1-hour data for average conditions in a season (Santiago) or month-grouping (Quito).

*Model constraints, boundary layer depth, and dilution constant*

We constrained the model with ozone, CO, and meteorological observations from both cities. VOCs (obtained as described in the previous section) were also used to constrain the model. NO and $NO_2$ were not constrained but the $NO_x$ sum was conserved (constrained). Following model recommendations, background concentrations were set to zero to avoid buildup of secondary species (Wolfe, 2023).

For boundary layer depth (PBLh) we used results from previous work done in Quito and Santiago. For Quito, we used the empirical model published by (Cazorla and Juncosa, 2018) that was inferred from balloon-borne measurements of PBLh and surface observations between 2014 and 2017. This approach was used to obtain PBLh in March 2021 and for the 2022 runs. For Santiago, we used results from the EMEP MSC-W model that is an offline chemical transport model (Simpson et al., 2012) as recently applied in Santiago and other South American cities (Pachón et al., 2024). For the seasonal runs in 2022, we scaled the March 2021 PBLh with an empirical factor using as reference previous work (Muñoz and Undurraga, 2010). Thus, we used 1/3 of the summertime values for winter PBLh and 2/3 for the fall. Fig. S10 shows PBLh for Quito and Santiago in March 2021. As stated in the F0AM model documentation, dilution is treated as a first order sink (Wolfe, 2023). Thus, we calculated a first order constant of dilution (kdil in the model) for each city by relating the time evolution of the PBL (m/s) to PBLh (m) at every step of the run. The bottom panels in Fig. S10 show kdil for both cities.

*VOC input*

For the VOC input, compounds were distributed among explicit VOCs available in the MCM using weighing factors indicated in Table S2 in the Supplement. For example, compounds measured as butanone/butanal were distributed as 50% butanone and 50% butanal. These correspond to MCM notations MEK and $C_3H_7CHO$, respectively. Thus, a total of 36 VOCs were input to the model resulting in 9189 reactions. Compound-grouping and notation used in this study, along with all input species are depicted in Table 2.

**Table 2.** VOCs measured in Santiago, Chile in March 2021 and used as model input.

| Group | Compounds |
|---|---|
| ALK | 1-butene, cis-butene, trans-butene, propene |
| ARO | benzene, toluene, styrene |
| | ethylbenzene, o-xylene, m-xylene, p-xylene |
| | propylbenzene, isopropylbenzene, 1,2,3-trimethylbenzene, 1,2,4-trimethylbenzene, 1,3,5-trimethylbenzene, 2-ethyltoluene, 3-ethyltoluene, 4-ethyltoluene |
| OXY | methanol, ethanol, phenol, cresol |
| ALD | acetaldehyde, acetic acid, glycolaldehyde |
| | acetone, propanal, methacrolein, methyl vynil ketone, butanone, butanal |
| ISO | isoprene |
| | alpha-pinene, beta-pinene, limonene |

*Frequencies of photolysis ozone column and albedo*

Frequencies of photolysis were modeled using the MCM option in the F0AM for both cities. In the Results section we compare the mean diurnal variation of $JNO_2$ and $JO^1D$ between both cities and with previous studies. For the ozone column and albedo,

we used 1-hour area-averaged MERRA-2 (Modern-Era Retrospective analysis for Research and Applications, Version 2) data
sets for Quito and Santiago (Global Modeling and Assimilation Office (GMAO), 2015a, b). This data selection was based on
previous work that showed good performance of MERRA-2 products for total column ozone in the region (Cazorla and Herrera,
2022). Table S3 contains a summary of options chosen in the F0AM for model runs.

**2.5.2 Sensitivity runs**

We ran a series of experiments to investigate the individual contribution of VOC groups to P(O$_3$) as well as the sensitivity of
P(O$_3$) to different combinations of VOC and NO$_x$ levels.

For determining the contribution of VOC groups to P(O$_3$), we ran the model starting with alkenes (ALK) and adding one VOC
group at the time until completion of all groups (ALK, ARO, OXY, ALD, and ISO). From these runs, we determined the
percentage contribution of each group to P(O$_3$) in each city. For the last group (ISO), the percentage contribution refers to
isoprene because adding monoterpenes essentially did not cause change.

We tested the sensitivity of P(O$_3$) to VOC levels P10, P50, and P90 (percentiles 10, 50 and 90 from MC simulations) under
observed NO$_x$ concentrations, 25% NO$_x$ reductions, and 75% NO$_x$ reductions in each city. VOC level P10 (P90) corresponds to a
30% decrease (increase) from P50 in Santiago and 40% in Quito. The group contribution runs were done using P50. In addition,
we performed seasonal runs in 2022 for the entire VOC range (P10, P50, and P90) under observed NO$_x$ levels in each city. Thus,
we present the results from a total of 50 model runs (25 per city).

**3 Results and discussion**

**3.1 March 2021**

**3.1.1 Air pollutant levels**

Ozone maxima in Santiago ranged between 30 to below 70 ppbv in March 2021, as shown by percentiles 10 and 90 of diurnal
profiles obtained with 1-hour observations in Fig. 2a. Looking at the entire data set, half of the days in the month had ozone
higher than 50 ppbv and there was a day with a maximum of 100 ppbv on March 4 (the school year started on March 1). Unlike
Santiago, Quito does not have a well-established ozone problem or season, despite equatorial solar radiation at high altitude and
urban emissions. Usually, 1-hour data remain below 50 ppbv. For example, in March 2021, ozone maxima during mostly sunny
days ranged between 30 to 50 ppbv for percentiles 10 and 90 as depicted in Fig. 2a. In previous years, high ozone in Quito (up to
ppbv) has been recorded episodically and has been associated with wildfires in the surrounding forests, usually during the
month of September (Cadena et al., 2021). During the study period, PM$_{2.5}$ levels (24-hour mean) were 17.6 µg m$^{-3}$ in Quito and
22 µg m$^{-3}$ in Santiago.

Regarding levels of NO$_x$, ambient concentrations of NO and NO$_2$ in Santiago in March 2021 were greater than in Quito by a
factor of 3 for the maximum at the 50$^{th}$ percentile, as depicted in Fig. 2b and 2c. However, levels of CO at the 50$^{th}$ percentile
from the mid-morning to noon were about similar (Fig. 2d), although data dispersion is within +/-25%. During the rush hours CO
can differ by +/-50%. The CO and NO diurnal profiles show strong traffic signatures in the morning and evening rush hours over
Quito. In contrast, the morning rush hour in Santiago stands out more, while the evening signature is delayed and less prominent.
These features are associated with the urban activity and work habits of citizens in both cities. In Santiago, work hours extend
into the evening and night, often past the work schedule, and citizens take a long time to return to their homes. In Quito, citizens
usually leave work before dark and usually do not extend the eight-hour work schedule.

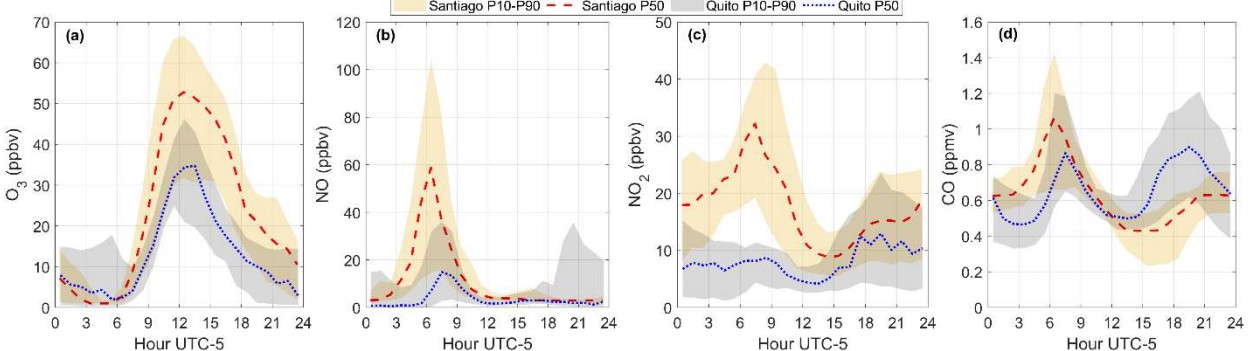

**Figure 2.** Diurnal variation of a) ozone, b) NO, c) NO$_2$, and d) CO observed in Santiago and Quito in March 2021. The red dashed line is the diurnal variation from observations in Santiago at the 50th percentile and the dotted blue line is the same but for Quito. The orange shadow is limited at the bottom and top by the 10th and 90th percentile diurnal variations from observations of every variable in Santiago. The gray shadow is the same but for Quito.

The diurnal NO$_2$/NO$_x$ ratio is an indicator of photochemical activity. It involves nitric oxide (NO) reacting with hydroperoxyl (HO$_2$) and alkyl peroxy (RO$_2$) radicals to form NO$_2$. The shape of the diurnal cycle of this ratio is similar between Quito and Santiago. However, Santiago's photochemical activity starts earlier than in Quito as given by an earlier increase of NO$_2$/NO$_x$ (Fig. 3) right after the rush hour dip (due to increased NO$_x$ from traffic emissions). This is related to the earlier start in traffic in Santiago as seen in NO and CO (Fig. 2b and d), which in turn leads to a distinct NO$_2$ morning maximum that is not present in Quito (Figure 2c). At noon and in the afternoon, NO$_2$/NO$_x$ at the 50th percentile (observations) is about similar in both cities, which indicates similar photochemical activity.

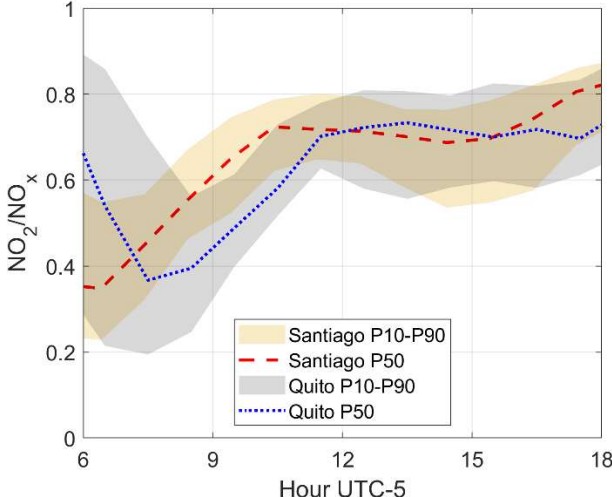

**Figure 3.** Diurnal variation of the NO$_2$/NO$_x$ ratio at the 50th percentile from observations in Santiago (red dashed line) and Quito (blue dotted line) in March 2021. Shadows depict limits at the 10th and 90th percentiles at Santiago (orange) and Quito (gray).

### 3.1.2 Model output

*JO¹D and JNO₂*

Photolysis reactions are key to radical and ozone production during daylight hours. Being Quito and Santiago located on the equator and in the subtropics, respectively, the magnitude of frequencies of photolysis are different due to the intensity of solar radiation at both locations. From model output, the diurnal variation at the 50th percentile of frequencies of photolysis of ozone (towards the production of $O^1D$) and $NO_2$ in March 2021 are presented in Fig. 4. As expected, both quantities are greater in Quito than in Santiago due to its equatorial location and high altitude. Previous work on the oxidation capacity of the Santiago air reported direct measurements of $JO^1D$ and $JNO_2$ taken during a field campaign in March 2005 (Elshorbany et al., 2009a). These measurements for diurnal maxima in Santiago were $JNO_2 \sim 0.008$ and $JO^1D \sim 2.5 \times 10^{-5}$ $s^{-1}$, which are similar to mean values obtained through modeling in this work.

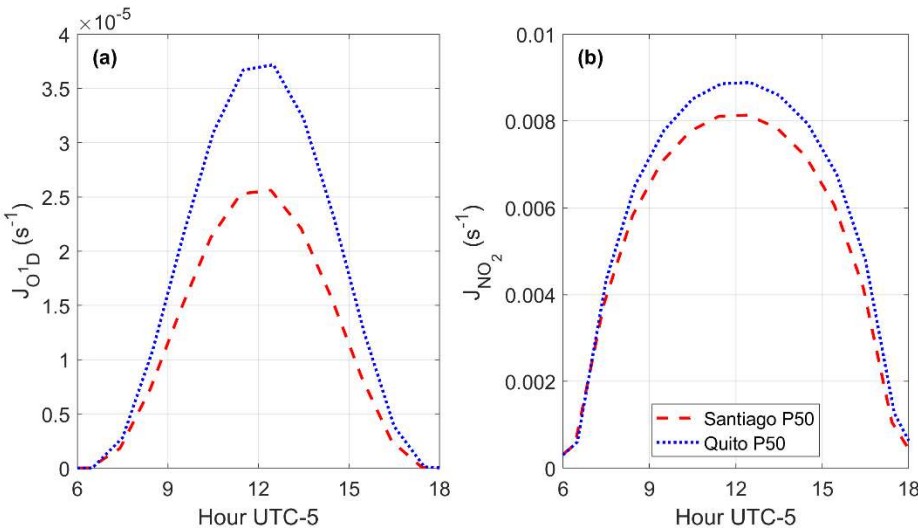

**Figure 4.** Comparison between the diurnal variation of frequencies of photolysis a) $JO^1D$ and b) $JNO_2$ in Quito and Santiago at the 50th percentile from model output.

*Ozone production and loss*

Ozone production rates were modeled for a range of VOC inputs and so are discussed in terms of ranges bounded by percentiles P10 and P90 obtained from the VOC distribution in each city (Fig. 5a). For Santiago, the range represents the uncertainty in the $P(O_3)$ simulation given the uncertainty in VOC measurements. Thus, the best estimation of $P(O_3)$ for Santiago is represented by P50 in Fig. 5a. The uncertainty was determined from the mean percentage difference of the upper and lower boundaries from percentile 50 and corresponds to +/- 32% (Fig. S11). For Quito, P10 and P90 set the boundaries of the possible range of $P(O_3)$ magnitudes expected for Quito calculated to the best of our knowledge from a distribution of VOC inputs. Due to a lack of in situ measurements we deal with the entire range when discussing Quito and we only present P50 curves in a referential way. The mean variability from percentile 50 is +/- 42% (Fig. S11).

The range of net ozone production rates at both cities at noon are comparable (22-45 ppbv $h^{-1}$), but in the mid-morning values could be higher for the upper boundary in Quito (up to 55 ppbv $h^{-1}$) (Fig. 5a). Further below we show that in Quito the $HO_2$ and

RO$_2$ abundance is greater when compared to Santiago (Fig. 9), but NO levels are lower (Fig. 2b). Thus, there is a compensating effect in the levels of HO$_2$ and NO in both cities in a way that application of equation (1) yields about similar ozone production rates. Previous work that determined net ozone production rates in Santiago from measurements and model calculations in 2005, reported substantially higher values for the mean P(O$_3$) diurnal maximum (160 ppbv h$^{-1}$) (Elshorbany et al., 2009b). Such high value is outside of our calculation range, although it is possible that after almost twenty years of reassessing P(O$_3$) in Santiago,

conditions changed. P(O$_3$) maxima presented in Fig. 5a for percentiles 50 of the distributions in both cities are consistent with values obtained in previous studies. For example, direct measurements of P(O$_3$) in Houston in 2009 were about 30 ppbv h$^{-1}$ on average (Cazorla et al., 2012). Another example is a recent sensitivity study of ozone production to NO$_x$ and VOCs in New York city that found P(O$_3$) within 26-37 ppbv h$^{-1}$ from modeling work with the F0AM (Sebol et al., 2024).

Losses to nitric acid are higher in Santiago than in Quito (Fig. 5b). This is consistent with higher NO$_x$ levels present in the

Santiago ambient air. Meanwhile, losses to hydrogen peroxide are higher in Quito, but the magnitude of this loss is one order of magnitude lower that the loss to nitric acid (Fig. 5c). Other losses are presented in Fig. S12 (Supplement). The loss due to ozone photolysis and the reaction of ozone with HO$_2$ are two and one orders of magnitude lower than the loss to nitric acid, respectively. The reaction of ozone with alkenes is also an order of magnitude lower than the loss to nitric acid, but this is higher than the loss to hydrogen peroxide. The upper boundaries of the loss to alkyl nitrates (P(RONO$_2$)) approach a digit and become

important along with the loss to nitric acid in both cities.

NO and NO$_2$ were unconstrained in the model, but the NO$_x$ sum was constrained. A comparison of measured and modeled NO/NO$_2$ ratio is presented in Fig. S13. From the mid-morning to the early afternoon values are about similar.

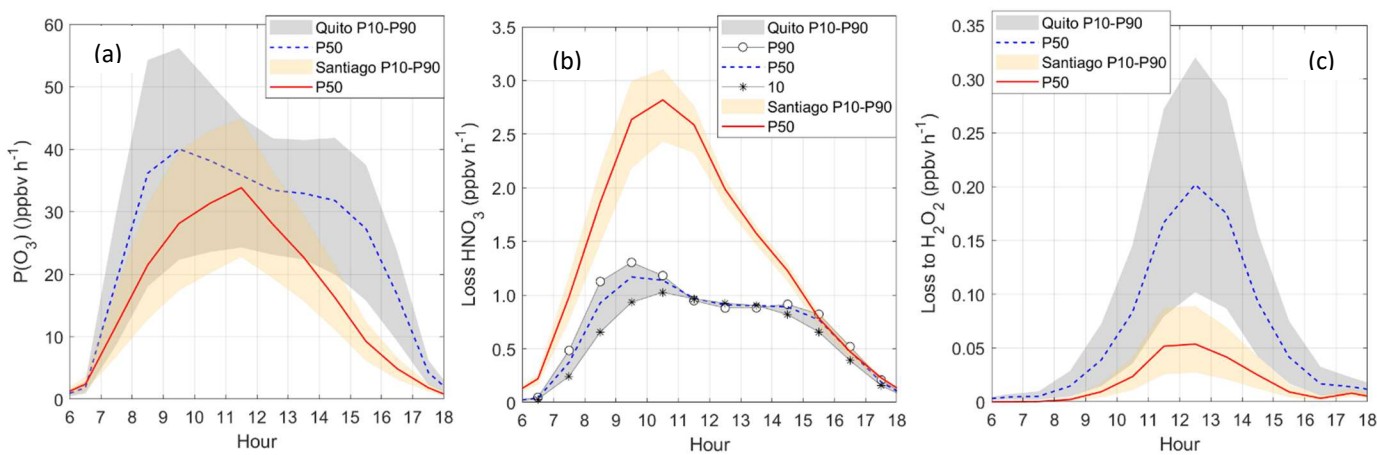

**Figure 5.** Diurnal variation of a) Net ozone production, b) Loss to nitric acid, and c) Loss to hydrogen peroxide in Quito and Santiago. Shadows correspond to boundaries obtained with VOC levels P10 (10$^{th}$ percentile) and P90 (90$^{th}$ percentile) from Monte Carlo simulations (orange for Santiago and gray for Quito). Symbols were added to better visualize percentile boundaries when needed. Results obtained at the VOC level P50 (50$^{th}$ percentile) are given by the solid red line for Santiago and blue dashed line for Quito in each graph.


An interesting point to investigate in the future is the connection between ozone production rates and ambient ozone, whose budget not only depends on atmospheric chemistry but also on meteorological and circulation factors, including long-range

transport. From our calculations, the range of P(O₃) in both cities is similar, but ambient ozone is higher in Santiago. This city has a year-round influence of the Subtropical Pacific High that determines a strong subsidence inversion, which is often

strengthened by low-level lows and occasionally disrupted by passing synoptic disturbances (Gallardo et al., 2002; Garreaud et al., 2002; Huneeus et al., 2006; Muñoz and Undurraga, 2010). Such meteorological condition is known to be the cause of pollutant accumulation in the boundary layer in Santiago. In contrast, Quito is not affected by a similar high-pressure system. In our simulations, we used information of the boundary layer evolution in both cities. The boundary layer in Quito can often grow up to 2200 m a.g.l. (Fig. S10) (Cazorla and Juncosa, 2018). Under such conditions, the first order dilution constant in the model

reached up to $2.5x10^{-4}$- $5.8x10^{-4}$ s$^{-1}$. Meanwhile in Santiago, the boundary layer can reach up to 1500 m and dilution constants usually stay below $2.5x10^{-4}$ s$^{-1}$ (Fig. S10). Additional research that uses a chemical transport model is needed to explain ambient ozone levels.

### *Ozone production regime*

The chemical regime of ozone production in Santiago is strongly VOC-limited as indicated by the ratio $LNO_x/(LNO_x+LRO_x)$ being very close to one for the entire range of the model output as depicted in Fig. 6a. This indicator is consistent with the formaldehyde to reactive nitrogen ratio ($HCHO/NO_y$) being well below 1 (Fig. 6b), and with the losses to nitric acid being substantial (Fig. 5b).

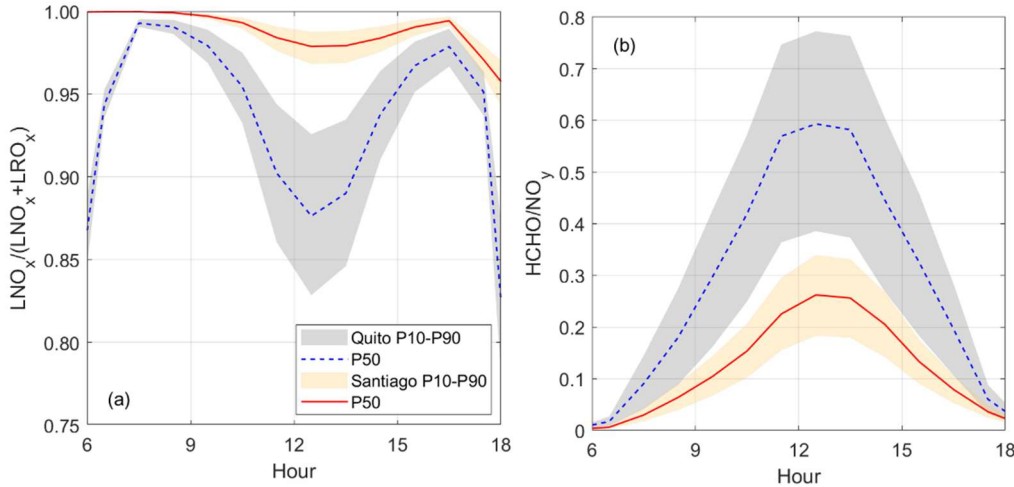


**Figure 6.** Indicators of the chemical regime of ozone production: a) Ratio $LNO_x/(LNO_x+LRO_x)$ (defined by Eq. 3 and 4), b) Ratio $HCHO/NO_y$ (formaldehyde to reactive nitrogen). Shaded area indicates model output boundaries obtained with the 10th and 90th percentiles (P10 and P90) of the VOC Monte Carlo distribution (orange for Santiago and gray for Quito). The solid red line and dashed blue lines correspond to model output at the 50th percentile (P50) for Santiago and Quito, respectively.


For the Quito runs, $LNO_x/(LNO_x+LRO_x)$ points to a VOC-limited regime as this indicator is above suggested thresholds of 0.5 (Kleinman, 2005c; Kleinman et al., 2001a) or revisited 0.74 (Schroeder et al., 2017). Meanwhile, the $HCHO/NO_y$ ratio is still below 1 for the entire range (VOC-limited), but the upper boundary approaches one digit. This indicates that with a higher VOC

content, the Quito environment is more prone to transitioning towards a more NO$_x$-limited regime. At the moment, a lack of in situ measurements constrains our ability to define how close the Quito environment is from the transition region. Even though from our calculations both cities are in the VOC-limited chemical regime, this character is noticeably more intense in Santiago from the indicators in Fig. 6. This finding illustrates that within the VOC-limited and NO$_x$-limited categories of ozone production there is a range of intensities that depend on the unique conditions that surround every urban area.

*VOC group contribution to P(O$_3$)*

The percentage contribution to P(O$_3$) caused by every chemical group (Table 2) during daylight hours in both cities is depicted in Fig. 7a (Quito) and 7b (Santiago). For the Quito simulations, alkenes and aromatics combined (blue and cyan in Fig. 7a) contribute 45% of ozone formation around the morning rush-hour, 70% at noon, and up to 90% in the afternoon. Meanwhile, the contribution of these two groups to P(O$_3$) in Santiago is under 50% throughout the day. The contribution of aldehydes and ketones is substantial over Santiago, which combined with alkenes and aromatics adds up to 80-90% of P(O$_3$) during daylight hours (magenta in Fig. 7b). In Quito, aldehydes and ketones are important in the morning, between 07h00-08h00, when the percentage contribution could be up to an additional 40-50%, while at noon and in the afternoon this contribution is about 20%. Isoprene contributes an additional 10-20% in Santiago and becomes increasingly important towards the afternoon. In Quito, we estimate that this contribution is about 10%. It must be noted that Central and Southern Chile suffered a longstanding drought between 2010 and 2022 (Garreaud et al., 2020) that likely caused an increase in isoprene emissions by vegetation due to hydrological stress (Jiang et al., 2018). The contribution of oxygenated compounds in both cities is marginal (black in Fig. 7). Future work needs to evaluate the contribution of alkanes in both cities as Liquid Petroleum Gas is used for heating and cooking (Chen et al., 2001).

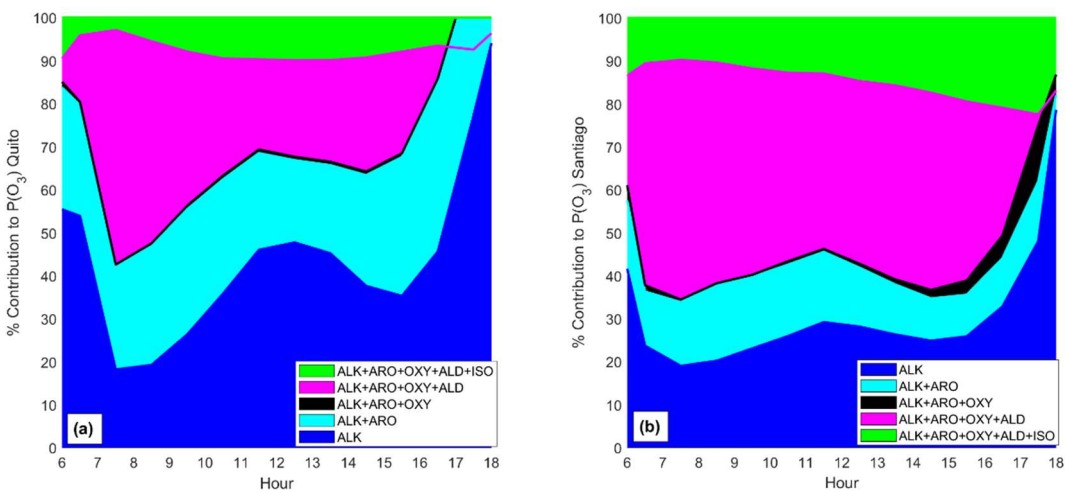

**Figure 7.** Percentage contribution to P(O$_3$) by chemical groups in a) Quito and b) Santiago obtained with model output at the 50$^{th}$ percentile from VOC Monte Carlo simulations.

 *Radical production and abundance*

The main sources of radical production are depicted in Fig. 8 and correspond to the photolysis of ozone followed by the reaction of $O^1D$ with water vapor (Fig. 8a), the photolysis of formaldehyde (Fig. 8b), the photolysis of HONO (Fig. 8c), and the reaction of ozone with alkenes (Fig. 8d). The first contribution has the largest magnitude in both cities. Computing this rate does not depend on VOC abundance for which percentile boundaries are absent in Fig. 8a. Thus, the curves depict the mean diurnal cycles whose maxima are about 1 pptv s$^{-1}$ in both cities, on average. However, within the month, variaibility in Santiago (0.5-1.75 pptv s$^{-1}$) is somewhat greater than in Quito (0.75-1.5 pptv s$^{-1}$). From meteorological data, the water vapor content at solar noon in Quito and Santiago is about similar (10 vs 9 g kg$^{-1}$) but solar radiation does not vary much on the equator while in Santiago daylight progressively lowers as the season advances towards the autumn.

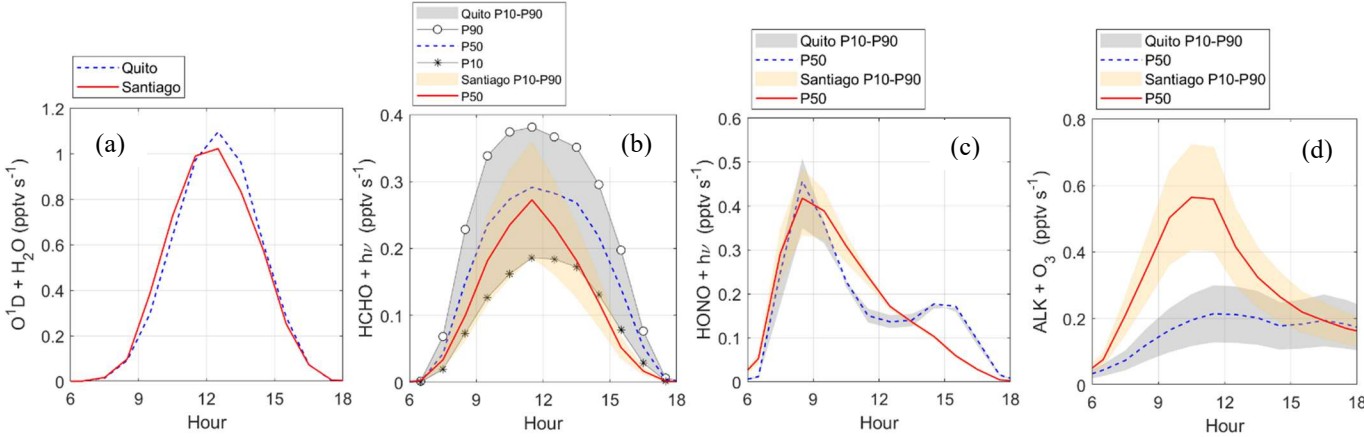

**Figure 8.** Diurnal variation of radical sources in Quito and Santiago: a) ozone photolysis followed by $O^1D + H_2O$, b) formaldehyde photolysis, c) HONO photolysis, and d) ozone reaction with alkenes. Shaded area indicates model output boundaries obtained with the 10$^{th}$ and 90$^{th}$ percentiles (P10 and P90) of the VOC Monte Carlo distribution (orange for Santiago and gray for Quito). Symbols were added to better visualize percentile boundaries when needed. The solid red line and dashed blue lines correspond to percentile 50 (P50) for Santiago and Quito, respectively.

P(HO$_x$) production from formaldehyde photolysis in Quito and Santiago is comparable (0.18-0.35 pptv s$^{-1}$) (Fig. 8b). Overall, this contribution is 2-3 times lower than the first source. In the morning, radical production from the photolysis of HONO is equally important in Santiago and in Quito (0.35-0.5 pptv s$^{-1}$), as indicated in Fig. 8c, which is expected given the high abundance of NO$_x$ in the morning. However, the contribution from this radical source gradually decreases from the mid-morning into the afternoon. The contribution from the reaction of alkenes and ozone is higher in Santiago (Fig. 8d), which is consistent with higher ozone levels. Previous work determined P(HO$_x$) from different sources in Santiago (Elshorbany et al., 2009a). They found rates of HO$_2$ production from formaldehyde photolysis of 0.15 ppt s$^{-1}$, on average, and rates of OH production from HONO photolysis of 0.4 ppt s$^{-1}$. Our calculations are consistent with these findings. However, rates of OH production reported by Eshorbany et. al. in 2009 from ozone photolysis are substantially lower (0.27 ppt s$^{-1}$, on average) . It is important to remark that this study only considers gas phase chemistry and urban emissions following previous work (Elshorbany et al., 2009b, a; Ren, 2003; Ren et al., 2013).

Radical abundances in both cities during the study time period are illustrated in Fig. 9. With higher frequencies of photolysis, the OH abundance in Quito for the entire simulation range (0.33-0.38 pptv) is overall greater than in Santiago (0.23 pptv) (Fig. 9a). The order of magnitude of radical abundances is similar to those found in other studies (Cazorla et al., 2021; Dusanter et al., 2009; Ren et al., 2013). Given the greater OH availability in Quito, reactions with VOCs yield $HO_2$ and $RO_2$ abundances that at noon in Quito are about three times greater than in Santiago (Fig. 9b and 9c). From the magnitudes in the latter figures, $HO_2$ and $RO_2$ are equallly important, which is consistent with previous work (Bottorff et al., 2023; Dusanter et al., 2009; Elshorbany et al., 2009a).

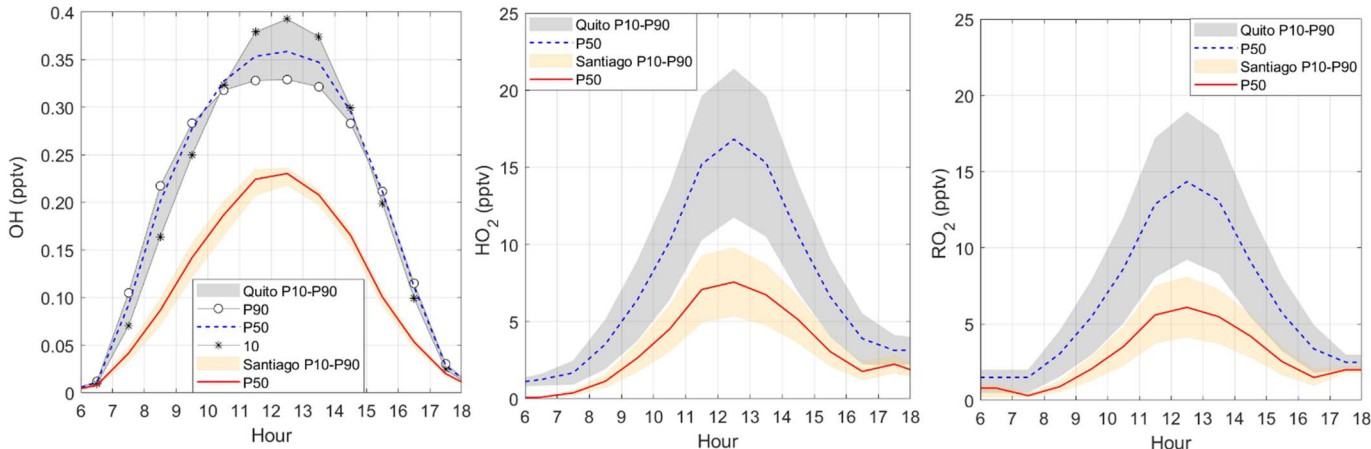

**Figure 9.** Diurnal variation of a) OH, b) $HO_2$, and $RO_2$ in Quito and Santiago. Shaded area indicates model output boundaries obtained with the 10th and 90th percentiles (P10 and P90) of the VOC Monte Carlo distribution (orange for Santiago and gray for Quito). Symbols were added to better visualize percentile boundaries when needed. The solid red line and dashed blue lines correspond to the output with the VOC 50th percentile (P50) for Santiago and Quito, respectively.

### 3.1.3 $NO_x$ and VOC scenarios

In this section we explore the effect on $P(O_3)$ due to changes in $NO_x$ and VOCs that could arrive from potential actions intended to improve air quality and help reduce SLCF. As stated before, diesel-based public transportation in South American cities is an important source of $PM_{2.5}$ and $NO_x$. It follows that a relevant scenario to consider is the effect on ozone production if this type of transportation were replaced, as it is often included in air quality and climate mitigation planning. Notice that Santiago's public transport bus fleet has roughly 10% of electric buses, a fraction that is expected to increase in the future. This type of action would significantly lower ambient $NO_x$ (and $PM_{2.5}$), although VOCs would also decrease. Another relevant scenario is exploring changes in VOCs. For example, applying changes in the quality of fuels to reduce the content of aromatic compounds would influence $P(O_3)$. Here, we explore some sensitivity scenarios, but additional research is needed to precisely determine the proportion in precursor reduction associated with specific environmental actions.

The effect on $P(O_3)$ with different combinations of $NO_x$ and VOCs is quantified in Fig. 10. The scenarios investigated were those of current $NO_x$ levels, $NO_x$ reductions by 25%, and $NO_x$ reductions by 75% at VOC levels P10, P50 and P90. For the latter, the percentage decrease (or increase) from P50 to P10 (or to P90) is 30% in Santiago and 40% in Quito. In Santiago and Quito, lowering the levels of VOCs would result in lower ozone production rates at all tested levels of $NO_x$ as presented by the

565 progression of panels in Figs. 10a to 10c (Quito) and 10d to 10f (Santiago). This feature is expected in VOC-limited environments. For example, at the current $NO_x$ Quito levels, reducing VOCs by 40% from the P50 level reduces the maximum $P(O_3)$ in 37%. In Santiago, a 30% decrease in VOCs would result in a 32% decrease in the maximum $P(O_3)$.

At lower levels of VOCs in Quito (P10), a 25% decrease in $NO_x$ causes a slight $P(O_3)$ increase in the morning and almost no change in the afternoon (Fig. 10a), while a 75% $NO_x$ reduction modestly raises the morning $P(O_3)$ but causes a drop at noon. As

VOC levels increase (P50 and P90), a 25% reduction in $NO_x$ causes a modest decrease in $P(O_3)$, but a 75% decrease causes the noon $P(O_3)$ to decrease by half. This result indicates that at higher VOC levels in Quito the ozone production chemistry is prone to transitioning towards a more $NO_x$-limited regime, especially at noon and in the afternoon. This aspect underscores the necessity of implementing in situ measurements of VOCs in Quito to accurately determine how close the Quito environment is to the transition region.

In Santiago, at the current levels of VOCs and $NO_x$ (P50), lowering $NO_x$ increases ozone production rates, especially in the morning and mainly when the decrease is substantial (75%), but the change at noon and in the afternoon is modest. Thus, at 9 am $P(O_3)$ grows from 25 ppbv $h^{-1}$ to 38 ppbv $h^{-1}$ (Fig. 10e). Similar effects are observed at the lowest (P10) and highest (P90) VOC levels (Figs. 10d and 10f). Thus, $P(O_3)$ increases in the morning and either there is a modest change or a decrease from noon to the afternoon. Therefore, $NO_x$ decreases cause the peak $P(O_3)$ to shift from noon to morning.

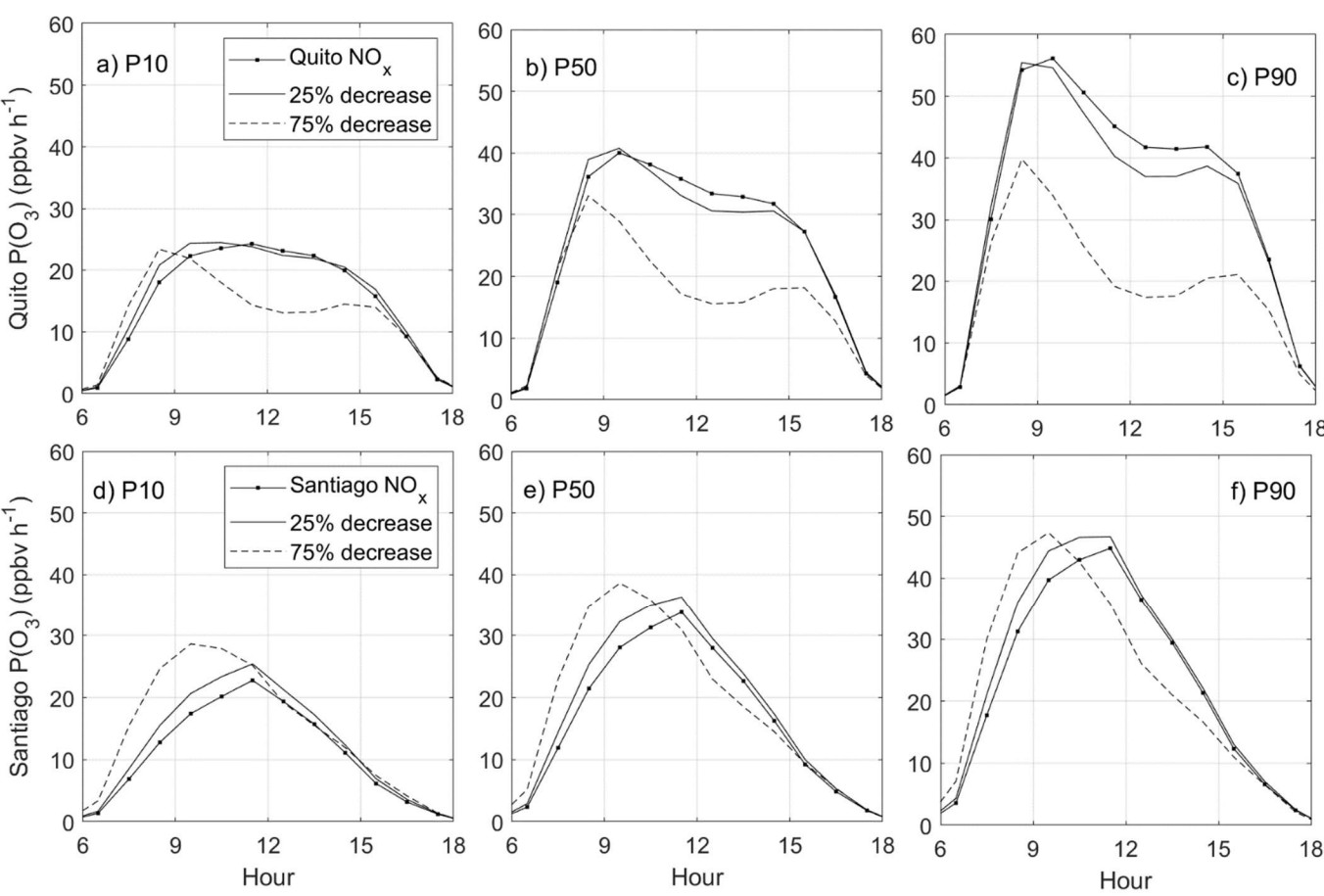

**Figure 10**. P(O₃) sensitivity to changes in NOₓ and VOCs in Quito (top panels) and Santiago (bottom panels) obtained with VOC levels P10, P50 and P90 from Monte Carlo simulations.

### 3.2 Mean conditions by season in 2022

As per additional conditions observed within a year in Quito and Santiago, Fig. 11 illustrates the range of mean levels of O₃, NO, NO₂, and CO that are typically observed in both cities (2022 1-h data). In Quito, the daily ozone maxima range from 30 to 45 ppbv, with higher ozone in September and lower in June-July (Fig. 11a). In contrast, Santiago data show clear seasonal differences in data distribution with mean summer peak ozone concentrations being more than 2.5 times higher than those of the winter (20 vs 55 ppbv) (Fig. 11b). The diurnal cycles shown in Fig. 11 are smoothed out when hourly data are averaged within a season. However, when inspecting the 1-hour time series in Quito, only on 1 day in 2022 ozone was higher than 60 ppbv. Meanwhile in Santiago, the peak ozone values were between 60-100 ppbv in 48 days. Table S4 shows statistics for 2022 data in both places.

Regarding NO, peak values in the diurnal cycle in Quito range between 30 to 55 ppbv, but there are times in the year when NO spikes substantially during the rush hour (Fig. 11c). In 2022, this spike happened in October-November. In Santiago, NO is substantially higher in the winter months, when in addition to transportation and industry, residential sources become significant (Fig. 11d). As temperatures are low, the mixing volume remains shallow and primary pollutants such as NO accumulate, while the actinic flux is not sufficient to activate photochemistry. In the 1-hour time series, 148 days in 2022 had NO higher than 100 ppbv in Santiago, while in Quito there were 31 days with morning NO of such magnitudes.

In Santiago, the winter-to-summer variability is also evident in the diurnal cycles of NO₂ and CO, with higher concentrations during the cold months (Fig. 11f and 11h). In contrast, the variability is small in Quito for NO₂, especially during daylight hours, and marginal for CO (Fig. 11e and 11g).

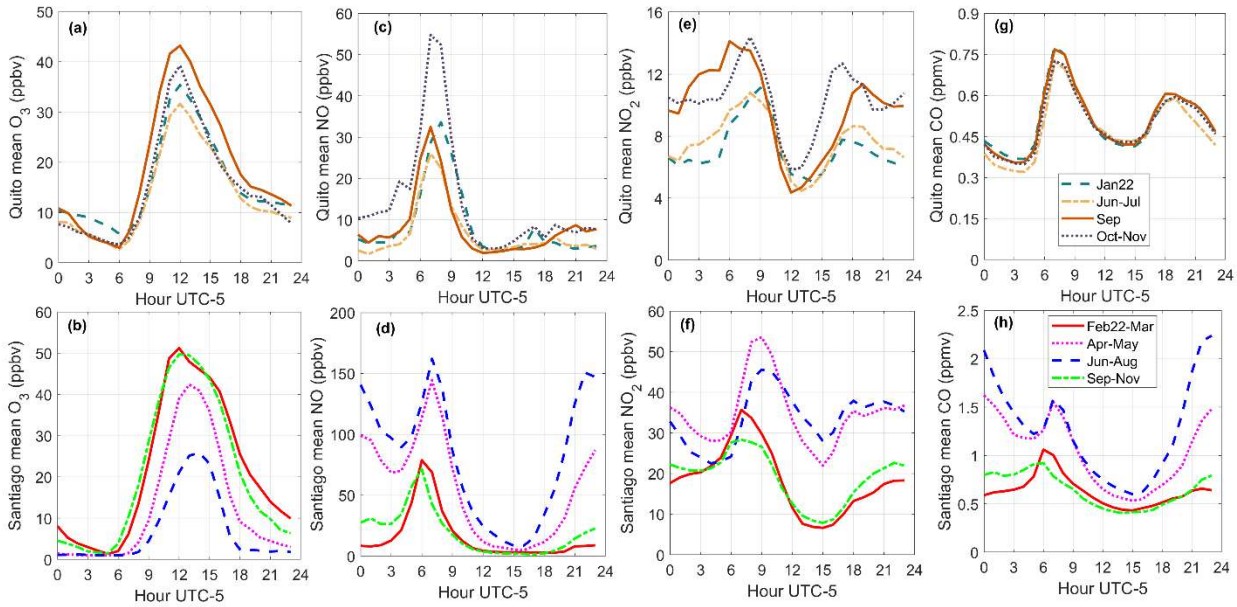

**Figure 11.** Mean air quality conditions observed within a year by season (Santiago) or month-grouping (Quito) presented comparatively for a) and b) ozone, c) and d) NO, e) and f) NO₂, and g) and h) CO.

With the range of conditions presented in Fig. 11, the range of ozone production rates calculated on average is depicted in the top panels of Fig. 12 for Quito and at the bottom for Santiago. In Quito, the mean ozone production rates roughly range from 15-40 ppbv h$^{-1}$ year-round (Fig. 12a to d), although in September the upper boundary is right under 50 ppbv h$^{-1}$. These magnitudes of P(O$_3$) resemble ozone production rates during the ozone season in Santiago (February-March, Fig. 12e). The rest of the seasons, P(O$_3$) in Santiago is lower by 10-20 ppbv h$^{-1}$. Therefore, with the range of VOCs used in the model, ozone production rates during sunny days in Quito stay high most of the year. In contrast, ozone in Santiago is high only during the ozone season. However, ozone levels in Quito are generally lower than in Santiago.

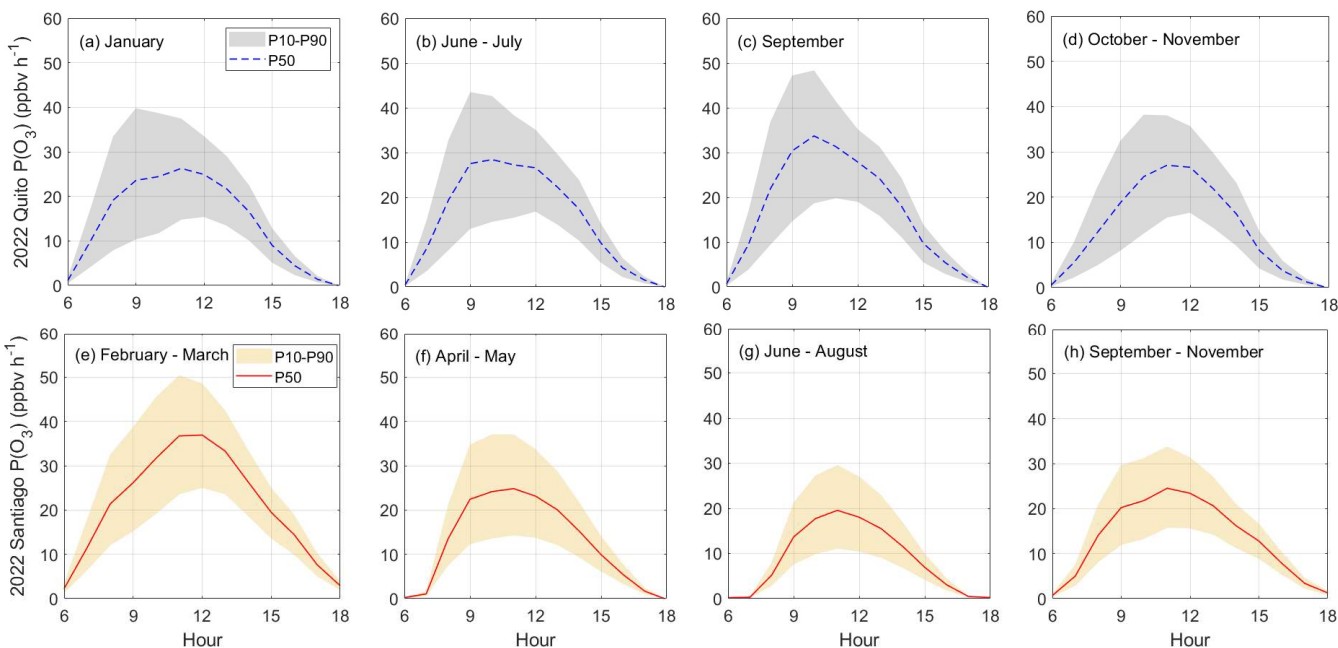

**Figure 12.** Comparison of the range of net ozone production rates observed in a year (2022) in Quito (top panels) and Santiago (bottom panels). Shaded area indicates model output boundaries obtained with the 10$^{th}$ and 90$^{th}$ percentiles (P10 and P90) of the VOC Monte Carlo distribution (orange for Santiago and gray for Quito). The solid red lines and dashed blue lines correspond to percentile 50 (P50) for Santiago and Quito, respectively.

**4 Conclusions**

Ozone production rates in Quito and Santiago (sunny days) were calculated using the F0AM (Framework for 0-D Atmospheric Modeling) for March 2021 and for mean conditions per season in 2022. The model was constrained with ground station observations (air quality and meteorology) and VOCs. The latter were measured in Santiago during part of March 2021. When measurements were not available, VOCs were generated with experimental VOC vs. CO linear regressions and Monte Carlo simulations within 35% of error in the observed values. We estimate an uncertainty of 32% in P(O$_3$) calculations for Santiago. Measurements of VOCs in Quito are not available for which we simulated the entire set of VOCs. To this end, we scaled Quito CO using Santiago linear regressions, but we perturbed these values within 50% using Monte Carlo simulations. The percentage variability in the range of calculated P(O$_3$) for Quito is 42% from the mean.

Santiago is impacted by a well-established ozone season during the austral summer, which extends into March. In March 2021, the $P(O_3)$ maximum was 32 ppbv h$^{-1}$ (+/- 32% or 22-42 ppbv h$^{-1}$). During this period, there were 11 days in Santiago when 1-hour ozone observations were between 60-80 ppbv and one day when ozone was 100 ppbv. In the summer 2022, the peak $P(O_3)$ was similar (37 ppbv h$^{-1}$ +/-32%), while the rest of the seasons $P(O_3)$ was 10-20 ppbv h$^{-1}$ lower. Meanwhile in Quito, our calculations indicate that the $P(O_3)$ range in March 2021 was 23-45 ppbv h$^{-1}$ at noon, but in the mid-morning the upper boundary could reach 55 ppbv h$^{-1}$. From calculations in 2022 we found a range between 15-40 ppbv h$^{-1}$ except in September when the upper boundary could reach 50 ppbv h$^{-1}$. However, ozone levels in Quito (1-hour) usually remain below 50 ppbv (only 1 day in 2022 was just over 60 ppbv). Therefore, we found that $P(O_3)$ in Quito is comparable or possibly even higher than in Santiago (during the ozone season), but ambient ozone is generally lower. The air quality in Santiago is permanently impacted by the Subtropical Pacific High, which causes a strong subsidence inversion. In contrast, Quito is not influenced by a similar high-pressure system. Furthermore, we found that often a deep boundary layer of up to 2200 m develops over Quito and the model first order dilution constant reaches $2.5 \times 10^{-4}$ - $5.8 \times 10^{-4}$ s$^{-1}$. Meanwhile in Santiago, we found that the boundary layer depth reaches 1500 m and the dilution constant stays below $2.5 \times 10^{-4}$ s$^{-1}$. These findings provide some indications of relevant physical factors but additional research using a chemical transport model is needed to explain ambient ozone levels and the ozone budget.

As per the chemical nature of VOCs that contribute the most to the formation of ozone, we found that alkenes in Quito contribute 45% in the morning and 70-90% at noon and in the afternoon. In Santiago, the contribution of alkenes and aromatics is about 50% throughout the day. Aldehydes and ketones contribute about 50% in the morning in Quito and 20% at noon and in the afternoon. In Santiago, aldehydes and ketones contribute an additional 40%, which adds up to 90% of the contribution along with alkenes and aromatics. We estimate that isoprene contributes about 10% in Quito and 20% in Santiago. The latter may be linked to the effects of a longstanding drought that affected Central and Southern Chile until 2022.

Ozone production rates of similar ranges in both cities result from a compensating effect in the magnitude of radical abundance relative to NO abundance. Thus, $HO_2$ and $RO_2$ radicals in Quito ($HO_2$: 12-22 pptv, $RO_2$: 9-18 pptv) are about twice the abundance in Santiago ($HO_2$: 6-9 pptv, $RO_2$: 4-8 pptv) but NO is two to three times higher in Santiago than in Quito. As in the case of ozone production rates, radical production rates from several sources are comparable in both cities. From ozone photolysis radical production rates are about 1 pptv s$^{-1}$; from formaldehyde photolysis 0.18-0.35 pptv s$^{-1}$; and from HONO photolysis 0.35-0.5 pptv s$^{-1}$. The contribution of ozone reactions with alkenes is greater in Santiago (0.4-0.7 pptv s$^{-1}$) than in Quito (0.12-0.3 pptv s$^{-1}$).

Indicators of the chemical regime of ozone production from model output, ($LNO_x/(LNO_x+LRO_x$ and $HCHO/NO_y$), point to Santiago being a strongly VOC-limited environment. This is confirmed by higher ozone losses to nitric acid than in Quito although losses to the production of organic nitrate are comparable in both cities. In addition, from sensitivity tests in Santiago, $NO_x$ reductions at all levels of VOCs lead to higher ozone production rates in the morning. In Quito, the two indicators also point to a chemical regime limited by VOCs. However, the value of $HCHO/NO_y$ (0.8) approaches 1 at higher percentiles of VOC abundance. At such VOC levels, $NO_x$ decreases cause a drop in ozone production especially at noon and in the afternoon, which indicates transitioning into a more $NO_x$-limited regime. However, a lack of VOC measurements in Quito limits our ability to better determine these thresholds. From sensitivity tests, VOC reductions lead to a general decrease in $P(O_3)$ in both cities.

Finally, our results remark the importance of implementing in situ measurements of VOCs in Quito and conducting more extended measurements in Santiago given the current conditions of environmental vulnerability and the need to implement policies that protect public health and climate.

**Appendix A: VOC measurements in Santiago, Chile**

VOCs were measured by a PTR-TOF-MS (Ionicon Analytik GmbH, Model 1000). Ambient air was drawn with an external pump (KNF, N86KN.18) through a sampling line (1/8''). The sample air was injected into the PTR from a T-union via a polyether-ether-ketone (PEEK) capillary (1/16'') conditioned at 80 ºC. The ion source was supplied with a 15 cm$^3$ min$^{-1}$ water vapor flow. The drift tube was operated at temperature, voltage and pressure of 353 K, 600 V and 2.3 mbar, respectively, corresponding to a reduced electric field strength (E/N ratio) of ~136 Td (Townsend) (1 Td =10-17V cm$^2$).

The mass scale was calibrated every 10 minutes using H$_3$O$^+$ isotope (21.0220), NO$^+$ (29.9970) and protonated acetone (59.0490), while the voltage of the multichannel plate detector was optimized weekly. The relative mass discrimination (transmission) was calculated before the campaign using the sensitivity and known reaction rates. The limit of detection (LoD) was calculated as three times the standard deviation of the background signal from zero air measurements (ENVEA, Model ZAG7001). We use a certified gas standard (Airgas Specialty Gases) valid until 2023, traceable to NIST (National Institute of Standards and Technology), for calibration checks for benzene (79.054), toluene (93.070), and ethylbenzene+xylenes (107.086). The error for toluene, ethylbenzene + xylenes and benzene was -1.2%, -34.6% and +27.2%, respectively.

PTR-TOF-MS provides several advantages, including mass range and high measurement frequency. However, this technique is susceptible to fragmentation that may interfere at a given m/z. The data used in this work resulted from a series of field campaigns aiming to optimize the measurements (e.g., E/N) and know the impact of fragmentation on benzene. We used specific ion peaks in the mass spectrum collected by (Pagonis et al., 2019) in a public online library to identify reasonable VOC candidates in urban environments. Table A1 shows the protonated parent molecules detected in Santiago's atmosphere and the compound assigned by the monoisotopic m/z for identification. In general terms, PTR has been reported as a reliable way to measure species with higher proton affinities, such as aromatics (Table A1). However, m/z 107.086 produces fragments that interfere with m/z 79.054 (assigned to benzene), and therefore, a conservative error of 35% should be considered for aromatics and up to 50% for small hydrocarbons without known molecular fragmentation. We also detected interferences with m/z 69.070 (assigned to isoprene) during the morning rush hours. On the other hand, reactive species such as acetaldehyde, with high levels in Santiago, should be interpreted considering the photochemistry of Santiago.

**Table A1**. Protonated parent molecules utilized in this study and VOCs assigned according to literature. More details can be found in Pagonis et al (2019) and in the PTR library. The table indicates names, parent ions and proton affinity obtained from https://webbook.nist.gov/chemistry/ and limit of detection.

| Compound assigned | m/z | Proton affinity (kJ/mol) | Limit of Detection (ppbv) |
|---|---|---|---|
| Methanol | 33.034 | 754.3 | 0.5 |
| Propene/Cyclopropane | 43.054 | 751.6/750.3 | 0.2 |
| Acetaldehyde | 45.033 | 768.5 | 0.4 |
| Ethanol | 47.049 | 779.4 | 0.5 |
| 1-Butene/2-Butene | 57.070 | ----- | 0.3 |
| Acetone/Propanal | 59.049 | 812/786 | 0.3 |
| Acetic acid/Glycolaldehyde | 61.028 | 783.7/N.A. [1] | 0.4 |
| Isoprene | 69.070 | 826.4 | 0.2 |

| | | | |
|---|---|---|---|
| Methacrolein/Methyl vinyl ketone | 71.049 | 808.7/834.7 | 0.2 |
| 2-Butanone/Butanal | 73.065 | 827.3/792.7 | 0.2 |
| Benzene | 79.054 | 750.4 | 0.09 |
| Toluene | 93.070 | 784.0 | 0.2 |
| Phenol | 95.049 | 817.3 | 0.2 |
| Styrene | 105.070 | 839.5 | 0.08 |
| Ethyl benzene/Xylenes | 107.086 | ----- | 0.09 |
| Cresol | 109.065 | N.A. | 0.1 |
| $C_9$-Aromatics | 121.101 | ----- | 0.05 |
| Monoterpenes | 137.132 | ----- | 0.08 |

[1]N.A.: Not Available

## Appendix B: Time series of VOCs measurements taken in Santiago

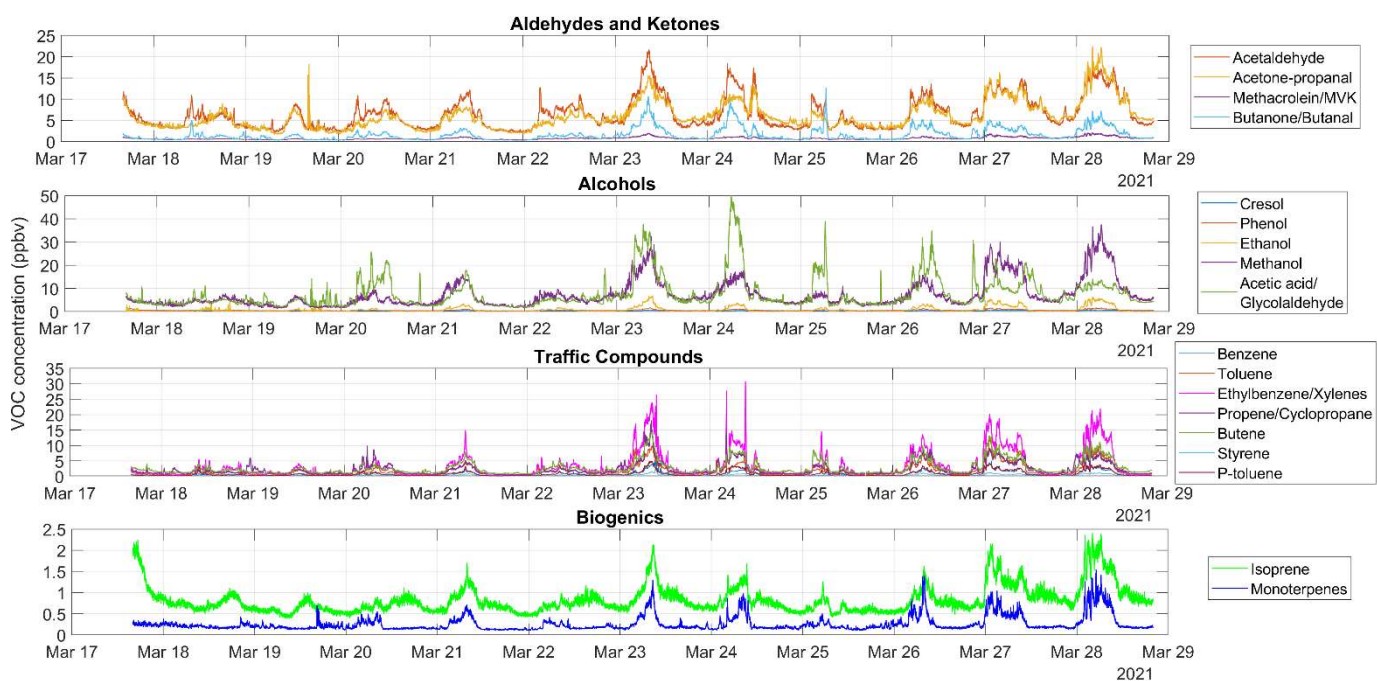

**Figure B1.** Time series (1-minute data) of VOCs measurements taken in Santiago, Chile from 17 to 28 March.

## Data Availability

Complete input data files (meteorology, air quality and VOCs) for the F0AM model base run (March 2021) can be accessed at:
Quito: https://data.mendeley.com/datasets/h4g7zfcj52/1
Santiago: https://data.mendeley.com/datasets/3cd2b8ktpz/1
Figures available at: https://observaciones-iia.usfq.edu.ec/ under the folder 'Photochemical_Model_Quito_Santiago'.
Quito public air quality data can be accessed at: https://aireambiente.quito.gob.ec/

Santiago public air quality data can be accessed at: https://sinca.mma.gob.cl/

MERRA-2 total column ozone and albedo data can be found at: https://giovanni.gsfc.nasa.gov/giovanni/

**Supplement**

Separate document

**Author contributions**

MC, LG, RS: conceptualization. MC: model methodology. RS: VOC methodology. MT: data curation and model runs. MC, MT: data analysis. MC: writing (original draft). All authors: writing, review and editing.

**Competing interests**

Authors declare not to have competing interests.

**Acknowledgements**

Thanks to Edgar Herrera and Lucas Castillo for their help at the beginning of this project. Thanks to Universidad San Francisco de Quito USFQ for their support during the development of this project. We acknowledge the use of public data from the Quito
Air Quality Network (Secretariat of the Environment, Quito, Ecuador) and the National Air Quality Information System (Ministry of Environment, Chile). Thanks to David Fahey from NOAA for his thoughtful advice and help. Thanks to Matthew Coggon from NOAA for his excellent input.

**Funding sources**

This research was supported by Universidad San Francisco de Quito USFQ through Collaboration Grant 2022-2023. LG and RS
acknowledge the support of FONDAP/ANID 1523A0002.

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
