# Peer review of "Comparative ozone production sensitivity to $NO_x$ and VOCs in Quito, Ecuador and Santiago, Chile"

_EGUsphere, 2024_

## Author Response (AR1)

**List of changes and corresponding sections:**

- Focus on model results and avoid strong policy statements: title, abstract, results, conclusions.
- Generate Monte Carlo simulations for VOCs for Quito and Santiago: Methods, Supplement, figures in supplement and in the link provided under Data Availability.
- Add figures for linear regressions VOC vs. CO: Supplement, link provided under Data Availability
- Add information about VOC measurements: Methods, Appendix A and B
- Rerun the model constraining $NO_x$ and unconstraining $NO$, $NO_2$: Methods
- Add PBLh and dilution constants to model runs: Methods, results, conclusions
- Sensitivity experiments VOC and NOx combinations: Methods, results, conclusions
- Add references: Introduction, References

Point-by-point responses to reviewers follow.

**Reviewer 1**

**First set of responses in the interactive discussion.**

**Second set of responses:**

**Please, find point-by-point responses (bold) below each comment (*Italic*).**

*I appreciate the authors' prompt answer to some of my concerns and the fact that they will take the errors of the VOC estimation into account for the next round of revision. Still, the fact that they use a different city environment (I mistyped New Mexico instead of Santiago) to establish the relationship between VOC and CO assumes that both cities' atmospheric conditions are interchangeable. To make the next revision smoother and shorter, I would like to elaborate more on a few things before the open discussion ends:*

**In the revised version of the paper, we generated a range of VOC inputs to the model using Monte Carlo simulations.**

*Here, I elaborate on the primary reason behind suggesting that faster/deeper vertical mixing of ozone over Quito is not a convincing explanation about why ozone levels are lower than Santiago: surface ozone concentration is a multifaceted parameter modulated by PO3, dry deposition rates, horizontal advection, horizontal diffusion, vertical advection (mainly through non-hydrostatic motions, which are masked in your analysis due to masking cloudy days), vertical diffusion, cloud chemistry, and background ozone values. The authors mentioned, "Additionally, previous studies demonstrate that on sunny days in Quito, a deep convective boundary layer develops in connection with thermal and mechanical eddies that break the early morning thermal inversion (Cazorla and Juncosa, 2018; Muñoz et al., 2023). Thus, we propose that strong convection at this tropical area helps mix and dilute ozone produced at the surface in the vertical direction." This part can be questioned in two aspects: i) rapid vertical diffusion known as "non-local motion" within the PBL only can ventilate surface concentration for a specie that decreases by the altitude such as NO2. Looking at Figure 5 (Carzola et al., 2021:*
*https://online.ucpress.edu/elementa/article/9/1/00019/117799/Characterizing-ozone-throughout-the-atmospheric), this is not the case for ozone. Therefore, a more rapid vertical mixing (expanded PBLH) should naturally enhance surface ozone (see Fig 6 and 7 in*
*https://www.sciencedirect.com/science/article/pii/S1352231010006187 as an example). This tendency is something that models always suggest (positive tendencies between surface ozone and vertical diffusion component) as long as there is no bizarre vertical ozone structure. If the authors meant deep convection (rapid vertical advection resulting from non-hydrostatic motions), that would bring up the fact that the presence of clouds could also reduce PO3. You either have clouds or don't consider them in the analysis. The most rigorous way of answering this critical question is to run a CTM model and carefully quantify the physiochemical processes responsible for shaping surface ozone concentration. Different air masses could be coming through the region (i.e., various weather patterns). Therefore, the authors either need to study these physiochemical processes in depth or tone down their explanation. Precisely, they should remove the negative effect of vertical mixing on surface ozone unless they can provide evidence that the ozone profile decreases by the altitude within the first 2-3 km.*

**The paper is about ozone production rates and the sensitivity to NOₓ and VOCs. Lines 440-464 in the original version refer to the interesting finding that ozone production rates in Quito are**

comparable to those in Santiago and yet ozone levels are lower. In the revised version of the paper, we reformulated this portion of the text to adjust the discussion only to the elements provided by our modeling work.

The several items indicated by the reviewer in the comment need to be demonstrated. Future research could address these and other questions which at the moment are outside of the main goal of the paper.

After revising the paper, to the best of our knowledge, we present the following findings:

- **Calculated ozone production rates in Quito with an array of VOC inputs to the model from Monte Carlo simulations were found to be as high or higher than in Santiago.**
- **Surface observations show that ozone in Santiago is higher.**
- **Santiago has a year-round influence of the Subtropical Pacific High that causes a strong subsidence inversion. Quito is not impacted by a similar system.**
- **Following suggestions by other reviewers, we included information about the boundary layer evolution. We found that the boundary layer in Quito is often deeper than in Santiago, and during those times the model 1$^{st}$ order dilution constant is higher.**
- **Additional research is needed in the future to explain ambient ozone levels and the ozone budget.**

*My issue with masking cloudy days (almost 11 days in March in Quito) is that the authors assume that the effect of clouds on surface ozone levels disappears after the sun is out, while we should recognize that ozone has a long lifetime and the resultant effect can linger for a prolonged time (depending on the wind condition). So, it is still important to discuss in the paper that 1/3 of March in Quito was cloudy, which could result in reduced background ozone levels for other days; however, it is very challenging to talk about ozone concentration without considering the transport component. That was my primary concern about using a 0-D modeling setup for a species constantly being transported in and out (0-D is perfectly fine to understand PO3, but it is unfit for ozone levels unless you add other elements such as Lagrangian or Eulerian transport components).*

**Masking is not part of the methods used in our paper. From the original to the revised versions of the paper we have presented in the most transparent way the exact conditions, dates, data sets, details, and methods under which the study was conducted. As explained in the first set of responses, our interest is to study photochemistry when the main condition is available, namely sunlight.**

*The last lingering issue is HONO. The fact that many chemical pathways outside the gas phase haven't been accounted for makes it challenging to study. In fact, the gas-phase production/loss (Case A in Figure 4 in https://linkinghub.elsevier.com/retrieve/pii/S0048969718329991) explains little about the diurnal behavior of HONO as the heterogeneous chemistry predominates.*
*I will look forward to seeing the revision!*

**The referred paper incorporates indoor emissions, soil emissions, and heterogeneous reactions as HONO sources to urban sources. For urban sources, they cite the work by Ren et. al. (2003) (https://doi.org/10.1016/S1352-2310(03)00459-X), which models HONO solely from gas phase chemistry using a box model. We used a similar method by the same author (Ren et al., 2013) to determine OH from HONO photolysis. In the revised version of the paper, we clarify that we**

**only use model HONO from urban sources and gas phase chemistry, and that other sources were not considered.**

**Reviewer 3**

**We thank Reviewer #3 for the useful suggestions.**
**Please, find point-by-point responses (bold) below each comment (*Italic*).**

*General Comments:*
*The paper by Cazorla et al. presents investigations of the ozone production regime and rates in Quito (city with population ~2.8 million; 2414 m above mean sea level; 0.19 S, 78.4 W) in Ecuador and Santiago (city with population ~ 8 million; 544 m above mean sea level; 33.5 S, 70.7 W) in Chile using the FOAM 0-D chemical model to suggest best measures for reduction of surface ozone pollution in the cities. The authors made efforts to use available measured datasets of VOCs and trace gases for the modeling analyses and found comparable ozone production rates of 15-35 ppb in both cities and conclude that both VOC and NOx control are essential for reduction of ozone, and that a focus on only NOx reduction would increase surface ozone in both cities. Data and detailed analyses of ozone formation chemistry from the South American region are limited to only a few sites and cities in the literature and thus, this study scores high on novelty for me as it is from an understudied region of the world. However as has been flagged by other reviewers, there are some major concerns that need to be addressed for readers to have sufficient confidence and clarity, before the manuscript can be recommended for publishing in ACP. I think it should be possible for the authors to address the concerns that have been raised earlier and in this review below. I suggest that the authors focus the revised MS as a study that can rather prescribe the most important measurement- modeling related gaps based on the limitations of the current measurement- modeling analyses for understanding ozone chemistry. Suggesting inferred tendencies for ozone formation chemistry, rather than definitive statements and toning down the strong policy related statements which are not supported by limitations in the present analyses, could help address many of the concerns. Having gone through the other review comments and the prompt replies by the authors (impressive!) and the manuscript, I shall limit my concerns/suggestions to points not covered in the other reviews and hope the authors find these useful. I look forward to the revised version!*

**Thank you. We substantially revised the manuscript following most of the suggestions.**

*Data AQ/QC; PTR-TOF-MS 1000 VOC data:*
*The authors state that they used a PTR-TOF-MS 1000 for VOC measurements and that these were available only from the Santiago site and only from 17-28 March 2021. While the operational settings of the instrument mentioned in the paper are standard, even for this period not enough QA/QC details have been provided. It would be good to have the transmission curve over the measured dynamic range and calibration curves for the compounds in the VOC std, even if this is only available for the period before and after deployment. Secondly, several compounds listed in Table 1 are known to require additional QA/QC for reliable quantification using the PTR technique. For example, formaldehyde is known to suffer from humidity artefacts and humidity based calibrations are necessary, else measured values can be gross underestimates. Secondly propene,*

*ethanol, acetic acid are also problematic because of fragments landing at these m/z of other compounds or some of the parent ion signal getting lost due to fragmentation. See for example Yuan et al., 2017 https://pubs.acs.org/doi/full/10.1021/acs.chemrev.7b00325 . It is not clear what measures/corrections were taken by the authors to ensure as far as possible that their compound attributions to the detected m/z are accurate. Authors should add these details for readers to have more confidence in the VOC measurements. There can be multiple contributors to the m/z the authors are currently attributing to single VOCs, and this needs to be corrected. See for example Table S1 of https://acp.copernicus.org/articles/24/13129/2024/acp-24-13129-2024-discussion.html )*

*Secondly, can the authors check if furan (detected at ~m/z 69.04) was also detected using the PTR-TOF-MS and show how the time series of furan and isoprene compare? Furan comes from combustion whereas isoprene comes from both biogenic and combustions sources in city environments (see for example https://acp.copernicus.org/articles/24/13129/2024/acp-24-13129-2024-discussion.html ). Authors could add local knowledge about biomass burning sources in Santiago from waste burning, open fires etc.. In Figure B1, isoprene never really goes below 0.5 ppb and this begs the question about whether it is purely biogenic. In other words, use of VOC tracers needs more caution if there are mixed sources.*

**The figure below shows the relative mass discrimination versus m/z. This transmission curve reflects the m/z dependency of our instrument. The transmission curve was made using a standard of benzene, toluene, ethylbenzene, xylene, chlorobenzene, trimethylbenzene, dichlorobenzene, trichlorobenzene. Note that every sensitivity (cps/ppbv) divided by the k rate constant ($10^{-9}$ cm$^3$ s$^{-1}$) was normalized to trichlorobenzene (39.6).**

[Figure]

$y = 0.0354 + 0.00526\ x$

**The figure below shows the calibration curves for benzene, toluene, and m/z 107,086.**

[Figure]

It is worth mentioning that we did not perform calibration curves for each of the reported VOCs. Instead, we relied on the reaction rate constant to yield a good approximation of VOC mixing ratios. Quantitatively speaking, the toluene-corrected signal improves by less than 1%, while for benzene, the uncorrected signal can be overestimated by up to 27%. These considerations should be kept in mind when interpreting VOC data.

We did not correct formaldehyde for humidity. Since humidity can produce up to 60% error in the formaldehyde measurement, we prefer to remove this species from the analysis.

Although the original objective of using VOCs in this work aims to perform a sensitivity study, it is important to acknowledge that (beyond the known BTEX fragmentation), we did not conduct specific studies for other species. Therefore, we suggest considering a 50% error without known molecular fragmentation.

In cases where we found multiple ion peaks in a nominal mass (e.g., toluene, acetaldehyde), we filter (deconvolution) the specific ion peak to avoid overestimating the calculated mixing ratios.

The resolution of our PTR does not permit us to clearly resolve isoprene from other compounds, such as furan, that can be present at m/z 69. In our experience, detecting Furan is only possible for us at high levels and when we suspect the occurrence of combustion processes.

Regarding the impact of combustion on isoprene levels, we have noticed a contribution in Santiago that is different from that attributable solely to biogenic emissions. Specifically, we found a morning spike of isoprene, which correlates with motor vehicle emissions. Although speculative because we do not have parallel isoprene measurements with GC, we hypothesize that vehicular sources emit a portion of isoprene, as reported in the literature. This is supported by measurements we made in closed parking lots for light-duty vehicles and buses, where we have measured high levels of isoprene. In any case, this analysis is out of the scope of this manuscript and is the subject of ongoing work.

We have no evidence of large-scale waste burning or fires during the measurement period. Therefore, the observation of an elevated background level of isoprene may be related to vehicle emissions.

*Comments related to the use of VOC/CO ERs:*

*Reviewer 1 has already flagged concerns about using VOC/CO ERs from Santiago for the Quito site and the authors have replied that the revised MS will employ a Monte-Carlo approach for VOC inputs. However even for Santiago, where VOC and CO measurements were available for some periods, it was not clear to me if the authors had used only nighttime data to determine these ratios and subtracted the background values for CO. Taking the mid-day CO median value could be one option for the determining the CO background for the Santiago data. Further a comparison of the ERs for toluene/CO, benzene/CO and sum of C8 aromatics/CO and sum of C9 aromatics/CO with those reported in other cities would be helpful. Please provide atleast in the supplement the plots for the VOC/CO ERs. Also what were the toluene/benzene ratios as per the measured time series? Please consider providing the seasonal ventilation coefficient data using the measured wind speeds and ERA 5 reanalyses boundary layer data for both sites in the supplement. This may also help to constrain the impact of dynamics in addition to chemistry at the two rather distinct sites.*

**Originally, we correlated the diurnal cycle of each VOC with the diurnal cycle of CO. Diurnal cycles were obtained overlapping data in a 24-hour plot and finding the hourly mean.**
**We checked these results following the suggestion. Thus, we correlated nighttime VOCs and CO (we subtracted the background CO as indicated). Both approaches yielded similar slopes, but the determination coefficients were generally lower for the second approach. However, the determination coefficients improved for cresol, acetaldehyde, acetone, and isoprene. Thus, only for the latter four compounds we used the linear fits obtained with the second approach. We updated Table 1 accordingly. Table S1 in the revised version contains the regressions obtained with both methods. Regression graphs for benzene and toluene can be found in Fig. S4 and S5. The entire compendium of regression figures (both methods) can be found in the link indicated in the Data Availability statement. We also expanded the discussion including a comparison of the ratios bezene/CO, toluene/CO, and c9aromartics/CO with those reported in the literature for other Latin American cities.**
**The following figure shows the benzene/toluene correlation:**

[Figure]

**We looked at the ERA5 data, but the values for boundary layer height (PBLh) were very different from previous work in which we measured (Quito) or modeled (Santiago) the PBLh (figures**

**further below). Thus, we did not use the ERA5 dataset but the outcome of other models. Please see under "***Comments related to the use of the box model:***".**

*The uncertainty is somewhat high for the VOC measurements and due to different chemical lifetimes and potentially different seasonal sources of both VOCs and CO, even for Santiago the extrapolation of the VOC/CO to infer VOCs in seasons other than summer (March) is not a logically sound proposition. The authors should acknowledge this but if they run the box model in a Monte Carlo method for a range of VOC levels also for Santiago, this will not necessarily undermine the box modeling study's usefulness.*

**We acknowledge the problem of applying the same ERs for Santiago during time periods when measurements were not available. We also acknowledge methodological limitations in measurements. For these reasons, we performed all the Santiago runs for a range of VOC concentrations determined by Monte Carlo simulations, as suggested.**

*Comments related to the use of the box model:*

*Several useful suggestions and limitations of the box model as a tool have already been flagged. While issues like transport and advection and dry deposition are typically missed, I am surprised that boundary layer variability and use of other ozone production regime proxies were apparently also not considered by the authors. See for example (https://www.sciencedirect.com/science/article/pii/S0045653521016568 )*

**We added PBL depths, please see below.**

*In order to avoid unrealistically high build-up of the model calculated long-lived secondary species, dilution of the model calculated species in the background air with zero mixing ratio should be considered according to Wolfe et al. (2016). Also as done in the above study (Kumar and Sinha, 2021), boundary layer height from ERA5 hourly reanalysis dataset (Hersbach et al., 2020) could be provided as a model input in the FOAM, which could also partially address concerns about the Quito site's dynamics.*

**We did set to zero the background mixing ratios to avoid build-up of secondary species. Additionally, it is important to mention that in the revised version of the paper we unconstrained NO and NO$_2$ and constrained NOx.**
**We used boundary layer heights (PBLh) from work done in previous studies. For Quito, we used an empirical model based on balloon-borne and surface measurements from Cazorla and Juncosa, 2018 (https://doi.org/10.1002/asl.829). For Santiago, we used data from modeling work done with the EMEP MSC-W model (Pachón et al., 2024) (https://doi.org/10.5194/gmd-17-7467-2024). Using PBL evolution and height, we calculated a dilution first order constant (kdil in the model, Fig S10). We included this information in the revised paper.**

**About ERA5, PBLh values are off when compared to work done in Quito and Santiago, so we did not use it. Please, see the following figures where we compare our data with ERA5.**

[Figure]

**Quito PBLh vs. ERA5**          **Santiago PBLh vs. ERA5**

*Finally, since the measured data used to constrain the box model are not comprehensive and have limitations in terms of seasonal and site coverage and species, I would advise employing more than one ozone production regime diagnostic proxy from the model analyses ( e.g. HCHO/NOy, $H_2O_2$/HNO₃ and O3/(NOy–NOx) ratios). While the study probes VOC and NOx limited regime indicators, using the VOC OH reactivity and NOx OH reactivity from the model, can help additionally to infer the intermediate VOC and NOx limited Ozone production regimes ( See for e.g. https://www.sciencedirect.com/science/article/pii/S0045653521016568 ). Here, the limited information on VOCs in the current study could still be a challenge, which the authors may want to acknowledge since measurements of many reactive alkanes and alkenes were not available at all. Finally use of the measured long term Ozone and NOx data in y Vs x plots for different seasons at both sites and ozone isopleths using hourly average daytime data (season-wise), could be worth examining for more insights concerning the conclusion of NOx reduction increasing ozone.*

**As suggested, we incorporated an additional indicator to improve this portion of the text. Thus, we used the HCHO/NO$_y$ ratio from the model in addition to the LNO$_x$/(LNO$_x$+LRO$_x$) ratio. Furthermore, on the paper we acknowledge the limitations of this discussion due to a lack of in situ measurements in Quito or sufficient measurements in Santiago. Regarding the rest of the suggested methods (5 additional suggestions), we do not think adding all sheds more light, especially when some of these methods rely on sufficient knowledge of measured compounds. It has been remarked that, given our limitations, the paper should not make strong policy statements. Having revised the paper, we agree. Ozone isopleths are tools commonly used for policy-making decisions, which at this point could be misleading especially to our community. For this reason, we think it is not prudent to include.**

*Specific comments:*
*Abstract: The current version is too policy prescriptive for the limitations in terms of the measured VOC dataset, NO2 measurements, assumptions about VOC mix and speciation. I would suggest that authors could focus this more on the box model results and leave out the policy prescriptive statements.*

**The revised version focuses primarily on the box model results.**

*L31: "it" is missing from combats climate change..*
**Fixed.**

*L36: "local make up" ..would chemical composition be a better choice here?*
**Changed.**

*L55 Ozone formation rather than ozone accumulation? Note you did not investigate meteorological drivers…*
**Fixed.**

*L78: VOCs and not VOCS*
**Fixed**

*Reactions R1 to R13:*
*Many of these are not chemically accurate…so please correct the schematic…*
*Oxygen is missing in R1*
*R5 would not proceed without M*
*R13: Neither balanced nor correct..carbon is missing…*
**Fixed.**

*L235: Please add transmission and calibration curves to the supplement*
**This is not a study meant to report precision VOC measurements for Santiago. Our aim is to study ozone production sensitivity to precursors, which we present within a range.**

*L240-255: Please add QA/QC for HCHO, ethanol and acetic acid atleast…*
**Formaldehyde not included. Please refer to response about VOCs.**

*L264: What is the basis for even 50-60% uncertainty…*
**This is a rough and conservative estimate based on the percentage difference in CO in both cities which ranges between +/-25% at noon and +/-50% during the rush hours.**

*Section 2.5.1:*
*I agree with Reviewer 1 about the fact that constraining O3, NO and NO2 as not being a good idea.. You could constrain O3 and NOx and let the NO/NO2 ratio vary in the model and then comparison with measured NO/NO2 and model NO/NO2 would actually be more meaningful and could add confidence to the model results if the comparison are good.*

**In the revised version of the paper, we rerun the model (all runs) without constraining NO and NO$_2$ but constraining the sum NO$_x$ as family conservation. We included a figure for the NO/NO2 comparison in Fig. S13.**

*Table 2: Isoprene cannot be considered as purely biogenic for the studied atmospheric environment given that night time levels seem to be above 500 ppt and correlation with other anthropogenic tracers like acetonitrile , furan and toluene (all of which can also be measured with the PTR-TOF-MS) have not been ruled out. Have you considered or discussed that isoprene could be coming from biomass burning too? This is important to clarify because in your study you emphasize isoprene as biogenic and its importance for the ozone control etc…*

**As explained earlier, we think isoprene has a component from traffic emissions. However, we do not have evidence of waste burning or fires during the measurement period.**

*L330: Please add the PM levels… also can you run the FOAM with a dry deposition proxy if the levels are high (say > 60 micorgram/ m3)*
**Added.**
**Levels are 17.6 μg m⁻³ in Quito and 22 in Santiago μg m⁻³ (24-h mean). We did not include dry deposition in the model.**

*L396: HCHO does seem to be really low…any comment?*
**We removed formaldehyde due to reasons explained earlier.**

L441-445: The box model cannot help with vertical mixing , right?
**We reformulated this portion.**

L500: ..Diesel based transportation: Actually lot of reactive VOCs are co-emitted too…so it will depend on absolute amounts of different fuels consumed by the fleet …please see for example https://www.sciencedirect.com/science/article/pii/S2590162121000186
**We reformulated this portion.**

*L599-605: After stating in the preceding section the serious limitations and uncertainties, I don't think one can make such strong policy prescriptive statements…please reconsider..*
**We reformulated this portion.**

*L617: Please mention manufacturer of the calibration gas standard and validity*
**Added.**

**Citation**: https://doi.org/10.5194/egusphere-2024-3720-RC4

**Response to Owen Cooper**

**We would like to thank Owen Cooper for his valuable comments. Below point-by-point responses (bold) to every comment (*Italic*).**

*January 31, 2025 Comments by Owen R. Cooper (TOAR Scientific Coordinator of the Community Special Issue) on: Comparative ozone production sensitivity to NOx and VOCs in Quito, Ecuador and Santiago, Chile: implications for control strategies in times of climate action*

*María Cazorla, Melissa Trujillo, Rodrigo Seguel, Laura Gallardo*

*EGUsphere [preprint], https://doi.org/10.5194/egusphere-2024-3720*

*Discussion started: 17 Dec 2024*

*Discussion closes: 1 Feb 2025*

*This review is by Owen Cooper, TOAR Scientific Coordinator of the TOAR-II Community Special Issue. I, or a member of the TOAR-II Steering Committee, will post comments on all papers submitted to the TOAR-II Community Special Issue, which is an inter-journal special issue accommodating submissions to six Copernicus journals: ACP (lead journal), AMT, GMD, ESSD, ASCMO and BG. The primary purpose of these reviews is to identify any discrepancies across the TOAR-II submissions, and to allow the author teams time to address the discrepancies. Additional comments may be included with the reviews. While O. Cooper and members of the TOAR Steering Committee may post open comments on papers submitted to the TOAR-II Community Special Issue, they are not involved with the decision to accept or reject a paper for publication, which is entirely handled by the journal's editorial team.*

*Comments regarding TOAR-II guidelines:*

*TOAR-II has produced two guidance documents to help authors develop their manuscripts so that results can be consistently compared across the wide range of studies that will be written for the TOAR-II Community Special Issue. Both guidance documents can be found on the TOAR-II webpage:*

*https://igacproject.org/activities/TOAR/TOAR-II*

*The TOAR-II Community Special Issue Guidelines: In the spirit of collaboration and to allow TOAR-II findings to be directly comparable across publications, the TOAR-II Steering Committee has issued this set of guidelines regarding style, units, plotting scales, regional and tropospheric column comparisons, and tropopause definitions.*

*The TOAR-II Recommendations for Statistical Analyses: The aim of this guidance note is to provide recommendations on best statistical practices and to ensure consistent communication of statistical analysis and associated uncertainty across TOAR publications. The scope includes approaches for reporting trends, a discussion of strengths and weaknesses of commonly used techniques, and calibrated language for the communication of uncertainty. Table 3 of the TOAR-II statistical guidelines provides calibrated language for describing trends and uncertainty, similar to the approach of IPCC, which allows trends to be discussed without having to use the problematic expression, "statistically significant".*

**Thank you. We checked and we believe our format complies with the guidelines.**

*General comments:*

*This paper provides useful analysis of the controlling factors of ozone production in Santiago and Quito and these findings are relevant to policymakers. Given the lack of VOC observations in Quito, I find the method for scaling the Santiago observations against Quito CO levels to be acceptable, and I agree with the advice provided by the editor that the study of Quito be treated as a series of sensitivity situations.*

**Thank you. We improved the scaling method through Monte Carlo simulations for VOCs. We present a sensitivity analysis, as suggested.**

*Line 448*

*The authors hypothesize that the diurnal variation of the boundary layer depth has a major impact on ozone levels in Quito, based on previous studies of boundary layer dynamics above Quito and Santiago. This hypothesis seems very reasonable and it would be helpful if the authors can provide some additional information on Quito's boundary layer dynamics, as provided by the cited studies. For example, what is the typical depth of the daytime boundary layer? Does it vary by season? Are there days when the boundary layer is capped, as experienced by Santiago? While further detailed studies are required to improve understanding of this phenomenon, I think the authors could quickly provide some basic analysis to demonstrate the timing of the boundary layer growth. Assuming water vapor measurements are available at the monitoring station, the authors can plot the diurnal variation of water vapor mixing ratio. As the boundary layer grows, dry air from aloft will be entrained to the surface, and the water vapor mixing ration will drop. Such a plot could be added to Figure 12.*

**Thank you for the useful suggestion. We do have information about the boundary layer depth (PBLh) and evolution in both cities. In the revised version of the paper, we included PBLh and first order dilution constants as model parametrizations and we discussed the differences. We believe this new information is insightful for which we did not add water vapor discussions.**

*Specific comments*

*line 27*

*Non-human populations: While there are plenty of studies that demonstrate the impact of ozone on human and health and vegetation, there isn't much at all on the impacts of ozone on the health of animals. Do the cited studies provide specific analysis of the impact of ozone on animals?*

**They mostly focus on the impact on human life and plants, so we modified these lines accordingly.**

*Introduction*

*When reviewing the impacts of ozone on human health, vegetation and climate, this would be an excellent opportunity to cite the work from the first phase of TOAR: Gaudel et al., 2018, Mills et al., 2018, and Fleming and Doherty et al. 2018.*

**Thank you. We included these useful references.**

*Line 63*

*It would be helpful to cite the recent TOAR-II paper by Putero et al. 2023. While ozone increased in many urban areas during the COVID-19 pandemic, it actually decreased in the free troposphere and at rural high elevations sites.*

**Included.**

*Line 525*

*Please be more specific regarding the climate benefits of PM2.5 reductions. IPCC AR6 concluded that reducing aerosols, which generally reflect sunlight (except for black carbon), would lead to a net warming (Szopa et al., 2021; Forster et al., 2024).*

**Included.**

*References*

*Fleming, Z. L., R. M. Doherty, et al. (2018), Tropospheric Ozone Assessment Report: Present-day ozone distribution and trends relevant to human health, Elem Sci Anth, 6(1):12, DOI: https://doi.org/10.1525/elementa.273*

*Forster, P. M., Smith, C., Walsh, T., Lamb, W. F., Lamboll, R., Hall, B., Hauser, M., Ribes, A., Rosen, D., Gillett, N. P., Palmer, M. D., Rogelj, J., von Schuckmann, K., Trewin, B., Allen, M., Andrew, R., Betts, R. A., Borger, A., Boyer, T., Broersma, J. A., Buontempo, C., Burgess, S., Cagnazzo, C., Cheng, L., Friedlingstein, P., Gettelman, A., Gütschow, J., Ishii, M., Jenkins, S., Lan, X., Morice, C., Mühle, J., Kadow, C., Kennedy, J., Killick, R. E., Krummel, P. B., Minx, J. C., Myhre, G., Naik, V., Peters, G. P., Pirani, A., Pongratz, J., Schleussner, C.-F., Seneviratne, S. I., Szopa, S., Thorne, P., Kovilakam, M. V. M., Majamäki, E., Jalkanen, J.-P., van Marle, M., Hoesly, R. M., Rohde, R., Schumacher, D., van der Werf, G., Vose, R., Zickfeld, K., Zhang, X., Masson-Delmotte, V., and Zhai, P.: Indicators of Global Climate Change 2023: annual update of key indicators of the state of the climate system and human influence, Earth Syst. Sci. Data, 16, 2625–2658, https://doi.org/10.5194/essd-16-2625-2024, 2024.*

*Gaudel, A., et al. (2018), Tropospheric Ozone Assessment Report: Present-day distribution and trends of tropospheric ozone relevant to climate and global atmospheric chemistry model evaluation, Elem. Sci. Anth., 6(1):39, DOI: https://doi.org/10.1525/elementa.291*

*Mills, G., et al. (2018), Tropospheric Ozone Assessment Report: Present-day tropospheric ozone distribution and trends relevant to vegetation, Elem. Sci. Anth., 6(1):47, DOI: https://doi.org/10.1525/elementa.302*

*Putero, D., Cristofanelli, P., Chang, K.-L., Dufour, G., Beachley, G., Couret, C., Effertz, P., Jaffe, D. A., Kubistin, D., Lynch, J., Petropavlovskikh, I., Puchalski, M., Sharac, T., Sive, B. C., Steinbacher, M., Torres, C., and Cooper, O. R. (2023), Fingerprints of the COVID-19 economic downturn and recovery on ozone anomalies at high-elevation sites in North America and western Europe, Atmos. Chem. Phys., 23, 15693–15709, https://doi.org/10.5194/acp-23-15693-2023*

*Szopa, S., V. Naik, B. Adhikary, P. Artaxo, T. Berntsen, W.D. Collins, S. Fuzzi, L. Gallardo, A. KiendlerScharr, Z. Klimont, H. Liao, N. Unger, and P. Zanis, 2021: Short-Lived Climate Forcers. In Climate Change 2021: The Physical Science Basis. Contribution of Working Group I to the Sixth Assessment Report of the Intergovernmental Panel on Climate Change [Masson-Delmotte, V., P. Zhai, A. Pirani, S.L. Connors, C. Péan, S. Berger, N. Caud, Y. Chen, L. Goldfarb, M.I. Gomis, M. Huang, K. Leitzell, E. Lonnoy, J.B.R. Matthews, T.K. Maycock, T. Waterfield, O. Yelekçi, R. Yu, and B. Zhou (eds.)]. Cambridge University Press, Cambridge, United Kingdom and New York, NY, USA, pp. 817–922, doi:10.1017/9781009157896.00*

**Response to Editor's comment:**

*I share the same concerns as the reviewer who posted on December 23, so I will not repeat those here.*

*Given that you simply do not have VOC data from Quito, I recommend that you instead treat your study of Quito as a series of sensitivity situations. For instance, you could vary the VOC mix to understand the sensitivity of PO3 to VOC classes. You may find little sensitivity to certain VOC classes and then you could identify potential emission sources (e.g., industry, automobiles, vegetation) for those VOC classes that do. That is, your current manuscript makes conclusions as if you actually had VOC data. I recommend that you change the focus of the paper to be on hypothetical scenarios. What VOC mix would allow you to actually impact ozone levels in Quito.*

**Thank you for the input.**

**In the revised version of the paper we present sensitivity experiments with a set of VOCs generated using Monte Carlo simulations and following suggestions by other reviewers.**

**Response to Reviewer 2**

*The editor's comment serves as the second review for this manuscript.*

**OK.**

---

## Author Response (AR2)

**Thank you for the comments. Below, point-by-point responses (bold).**

The authors put a great deal of effort into revising the paper. They recalculated everything more stochastic to provide a broader range of possible outcomes.

Here are a few more comments to consider:

Figure 5. The plot layouts aren't consistent. For instance, the middle panel includes other percentiles.

**All figures contain only percentiles 10, 50 and 90. Some contours are hard to see, so we just added symbols to better visualize the percentile contours when needed. We added this clarification in the captions of Figures 5, 8, and 9.**

Line 455-465. This part of the discussion is oblivious to the vertical mixing mechanism. A more expanded PBLH should indeed dilute the ozone concentration only if we assume that there is no non-local vertical mixing between upper PBLH with air parcels close to the surface (a subsidence compensation for the rapid vertical mixing, see Figure 1 in https://www.atmos.albany.edu/facstaff/rfovell/NWP/pleim-2007.pdf). If the concentration of our targeted compound decreases with altitude, this way of mixing (which is faster when surface kinetic heat fluxes are large, resulting in expanded PBLH) will further dilute the concentrations because high concentrations of the species near the surface will be rapidly exchanged by lower values aloft. This condition can be completely reversed for a species whose concentrations generally increase by altitude. Perhaps, when surface ozone could reach 200 ppbv near the surface, the vertical profile of ozone during these events followed the first example, but given the fact that ozone levels are much lower in your case and your ozonesondes observations indicate higher ozone aloft, an expanded PBLH should lead to more ozone, fully overpowering the dilution effect. If the authors want to discuss the relationship between ozone concentration and meteorology, they need to be very thorough and not pick and choose the part of the picture to partly validate their claims. There are also no discussions about the regional background of ozone and wind patterns. These cities may receive different air masses with varying background ozone concentrations during the season. Dry deposition is another component to consider. Maybe some discussions about LAI or varying surface resistance between two cities can help (https://acp.copernicus.org/articles/19/14365/2019/acp-19-14365-2019.html).

**This comment contrasts with the comments we received from two other reviewers who considered it relevant to include a comparison between the depth of the boundary layer and boundary layer evolution between the two cities to explain differences in dilution conditions (please see comments by Reviewer2 and Owen Cooper in the interactive discussion). We followed these valuable suggestions, and we found that often the boundary layer in Quito is deeper and the first order dilution constant is faster than in Santiago. We included this insightful information to strengthen the discussion, as requested.**

**In this paragraph we do not show any figure, but the reviewer mentions that we make claims based on a picture (number not specified) with ozonesonde observations. This is not accurate**

because we do not present such information in this paragraph. Perhaps the reviewer is referring to the previous version of the paper in which we described the shape of the ozone profiles collected in Quito (i. e., https://doi.org/10.1002/asl.829). In this regard, it seems that the reviewer only focused on the shape of the 7 am profiles, which are not representative of the noon and afternoon profiles that show mixing when the boundary layer has reached its maximum depth. In the early morning, ozone becomes titrated by traffic NO during the morning rush hour. Under these conditions, of course ozone gives the impression of being lower at the surface and higher aloft, but this is just an effect of the morning ozone titration with NO. If you look at the noon and early afternoon profiles in the same paper, ozone is lower at higher altitudes. Thus, the argument presented by the reviewer in the sense that lower ozone aloft mixes with higher ozone at the surface, precisely holds for the ozone profiles we have observed at noon in Quito. In any case, we do not discuss ozonesonde morphology in this paper. Also, the surface ozone value mentioned by the reviewer (200 ppbv) is completely out of range for Quito. Please refer to the air quality discussion on the present paper.

Regarding the regional background of ozone, we do not address this aspect in this paper. As indicated before, this study deals exclusively with the outcome from a photochemical box model.  A regional/urban scale model that allows for a detailed inclusion of boundary conditions or upwind precursor sources needs to be performed in the future to assess transport and background ozone effects.

The reviewer also mentions dry deposition. As stated in the methods, we only considered chemical sources and sinks of ozone. In any case, it is important to realize what the order of magnitude of this physical sink is with respect to the order of magnitude of the chemical production of ozone. For example, for a typical ozone concentration in Quito at noon of 50 ppbv and boundary layer height of 1500 m, with deposition velocity for ozone of 0.4 cm $s^{-1}$ (Seinfeld and Pandis, 2016), the dry deposition is 0.48 ppb/h. This quantity is two orders of magnitude lower than the chemical production of ozone found in our study.

Having analyzed carefully this comment, we do not find a solid reason to make modifications to this portion of the text.

L 470. Two indicators aren't consistent for Quito for a less photochemically active environment. I suggest including only times when enough photochemistry is involved to have meaningful discussions about the chemical conditions. In light-limited regimes, it doesn't matter what these ratios are because the response of PO3 to its precursors becomes muted. Maybe you should cut the x-axis to 10 AM to 4 PM.

It is hard to interpret what exactly the reviewer means by inconsistent indicators for "less photochemically active environments". There is also mention of "light-limited regimes". It seems that the reviewer is under the impression that there is not enough sunlight in equatorial Quito. As demonstrated in the previous set of answers, this is not the case for this city located right on the equator at high altitude. In addition, we only deal with photochemistry under sunny conditions in both cities. Please refer to the first set of answers in the interactive discussion and to the Methods.

According to the indicator thresholds described in the text (references given), both figures, 6a and 6b, show that the regime is VOC-limited for the model output obtained with the range of VOCs used for Quito. The reviewer suggests modifying the axis and only showing the output between 10 am and 4 pm. However, this range is already visible in the current figures, making this modification unsubstantial.